# Experimentation for Different Scheduling Policies on Queues: Mixed Differences-in-Q Estimators Based on Little's Law

## Abstract

In data centers, tasks are dispatched to various servers to evenly distribute the workload. When a data center considers implementing a new scheduling algorithm, it typically conducts an A/B test prior to deployment to assess the real-world impact of this new method. However, a straightforward A/B test might be interfered with so-called "Markovian" interference. We utilized the Differences-in-Q estimator, as developed by Farias et al. (2022), and introduced mixed Differences-in-Q estimators grounded in Little's Law. We show that our A/B testing methods significantly reduce bias and variance when testing various scheduling policies. Extensive simulations were conducted under scenarios like non-stationary arrival rates, heterogeneous service rates, and communication delays. These simulations highlight the robustness and efficacy of our A/B testing approach.

## 1 Introduction

In the era of the digital economy and Generative AI, the surge in reliance on technological solutions has been unprecedented, which necessitates the deployment of extensive computational resources across global data centers. For example, in managing the colossal volume of billions of requests daily, Google employs sophisticated scheduling algorithms to efficiently dispatch these queries across a global network of servers (Abts et al., 2010; Brewer et al., 2016). Similarly, platforms like ChatGPT process billions of prompts through extensive models, leveraging advanced scheduling and computational techniques to generate tokens (Stojkovic et al., 2024).

Against this backdrop, the role of online scheduling systems becomes paramount due to the need to balance response time with computational costs under a myriad of constraints. These policies must account for geographical location to minimize latency, the computational load to optimize resource usage, and maintenance schedules to ensure system reliability without compromising service quality. Additionally, they must handle the heterogeneity of hardware resources, which can vary widely in capability and availability across different nodes. Furthermore, these scheduling decisions are continuously adjusted in response to fluctuating demand. Those practical constraints make the design and implementation of effective scheduling policies a complex yet critical task for maintaining the high performance and scalability of computational infrastructures.

Therefore, to effectively develop a new scheduling policy, relying solely on theoretical analysis under restrictive assumptions is insufficient. In industrial practice, the real performance of any new policies must first be validated through A/B tests. In an A/B test, tasks (requests, prompts) are randomly assigned to a treatment and a control group with different scheduling policies. Metrics such as the average response time between tasks in the two groups are then compared to measure the effectiveness of the newly developed policies.

However, such A/B tests may produce biased results due to interference between different scheduling policies. One scheduling policy may influence the server load, which in turn affects the response time of tasks governed by another scheduling policy.

Such interference is well-documented in the literature and is termed intertemporal interference or, when underlying states and Markovian structure are present, Markovian interference. To address this issue, Farias et al. (2022) propose an innovative Differences-in-Q (DQ) estimator. The DQ estimator

works by summing all subsequent rewards to estimate Q-values and then using the difference in Q-values to estimate the global treatment effect. This approach can substantially reduce the bias arising from the evolution of the underlying state. The effectiveness of the estimator is further validated through a Douyin simulation study (Farias et al., 2023), which demonstrates significant bias reduction. However, the method may still suffer from high variance.

Therefore, to address the high-variance issue of the DQ estimator, we propose a mixed Differences-in-Q estimator for Bernoulli trials, which generalizes the Differences-in-Q estimator of Farias et al. (2022). Our approach leverages Little's Law (Little, 2011), a fundamental principle that virtually applies to any queuing system. Little's Law states that, in a stable system, the long-term average queue length (include tasks in service) equals the average arrival rate multiplied by the average response time (waiting time + service time). Specifically, we maintain two Q estimators: one for the average response time and one for the average queue length. By Little's Law, their ratio gives the long-term average effective arrival rate. We then compute optimal mixing weights and combine the two Q estimators to reduce variance.

Our contributions are summarized as follows:

1. We develop a model of A/B testing in data centers within the framework of the supermarket model (Mitzenmacher, 2001), where incoming tasks are randomly assigned to servers according to different load balancing policies. This formulation enables us to rigorously capture the performance impact of alternative scheduling strategies and provides a natural setting for evaluating treatment effects under randomized experiments. Although we focus on the supermarket model purely for clarity of exposition, the underlying methodology is broadly applicable and can be extended to essentially all queueing network settings.

2. We propose a mixed Differences-in-Q estimator that leverages the unique features of the queueing model. Our method is easy to implement, and we prove that this estimator can achieve low bias and variance compared to other estimators.

3. We conduct thorough simulations in various practical contexts, including heterogeneous service rates, non-stationary arrival rates and communication delays. We also test many different scheduling policies. Our numerical results demonstrate that our A/B tests can achieve low bias and variance across a wide range of practical scenarios.

Finally, our findings highlight a broader insight: incorporating domain knowledge—specifically, the structural properties of queueing systems—can substantially reduce variance and improve the reliability of performance evaluation.

## 2 LITERATURE REVIEW

**Little's Law.** Little's Law, first proposed in Cobham (1954); Morse (2004) and formally proved in Little (1961), has been widely studied across different settings and applications. In its standard form, it states that the average queue length in a stable queueing system, equals the product of the average arrival rate and the average response time. Early studies considered parallel-server systems that were empty at both $t = 0$ and $t = T$. A typical example is a supermarket that opens and closes with no customers. Later, the law was generalized to allow nonzero initial and terminal queue lengths, with the average arrival rate redefined to include tasks already present at $t = 0$. Further details are provided in Appendix A.

**Multi-server scheduling polices.** There are many existing load balancing policies that enjoy good theoretical properties. The join-the-shortest-queue (JSQ) policy routes tasks to the least-loaded server and has strong theoretical foundations (Mukherjee et al., 2016a; Zhao et al., 2021; Banerjee and Mukherjee, 2019; 2020). Its drawback is the need for global queue-length information, which causes high communication cost. To address this, the power-of-$d$ policy samples $d \geq 2$ servers and assigns tasks to the shortest server. Power-of-2 achieves exponential gains over random assignment (Mitzenmacher, 2001). Later studies proved mean-field convergence (Graham, 2000), heavy-traffic limits (Mukherjee et al., 2018b; 2020), and asymptotic optimality of the power-of-$d$ family. Extensions also analyze networked servers with graph topologies Mukherjee et al. (2018a); Budhiraja et al. (2019); Rutten and Mukherjee (2023a;b). Other policies include MJSQ-$r$, which mixes random and shortest-queue routing (Banerjee et al., 2023). The join-the-idle-queue (JIQ) assigns jobs

to idle servers, or randomly otherwise (Badonnel and Burgess, 2008; Lu et al., 2011; Stolyar, 2015). JIQ-$d$ policy generalizes this by combining idle assignment with power-of-$d$ routing (Mukherjee et al., 2016b).

**Experimental design under intertemporal interference.** Intertemporal interference occurs when a previous treatment affects outcomes in subsequent periods. To address intertemporal interference, switchback experiments are commonly employed and have been widely studied in the literature. In these experiments, the experimenter alternates between treatment and control, typically on an hourly basis. Under Markovian assumptions, optimal switchback designs have been thoroughly analyzed by Glynn et al. (2020), Hu and Wager (2022), Li et al. (2023), and Jia et al. (2023). Beyond Markovian settings, Bojinov et al. (2023), Hu and Wager (2022), Xiong et al. (2023a), Xiong et al. (2023b), Basse et al. (2023), Xiong et al. (2019), and Ni et al. (2023) have investigated various aspects of switchback experiments. In the data center environment, the task arrivals are highly variable, leading to high variance in the target metrics due to factors like time-of-day effects. As a result, the variance of switchback estimates can be substantial relative to the treatment effect.

## 3 PROBLEM SETTINGS

In this paper, we consider a queueing network with $N$ parallel servers: tasks arrive as a Poisson stream with rate $\lambda N$, as depicted in Figure 1. The arrival times are denoted by a sequence $\{t_j : 0 = t_0 < t_1 < \cdots\}$. When a task arrives, the system assigns it to one of the servers according to a scheduling policy. Tasks are served according to the first-in-first-out protocol, and the service times for tasks in the $i$-th server follow independent exponential distributions with rate $\mu_i$. Temporarily, we set $\mu_1 = \mu_2 = \cdots = \mu_N = 1$ and leave the discussion of heterogeneous service rate settings in Section 5 later.

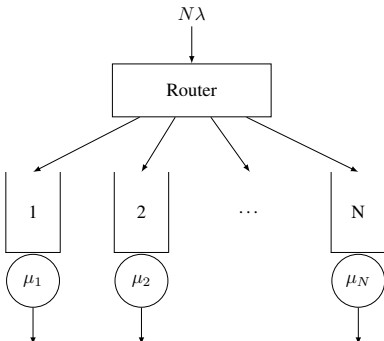

Figure 1: The parallel queue service system

For the ease of exposition, we now introduce some notations. We define $q_i$ to be the queue length (including the task in service) in the $i$-th server and $s_i$ to be the proportion of servers with at least $i$ pending tasks. Equivalently, we denote

$$s_i = \frac{\sum_{j=1}^{N} \mathbf{1}_{\{q_j \geq i\}}}{N} \quad i = 0, 1, 2, \cdots. \tag{1}$$

The state space of this parallel-server system is given by an infinite dimensional vector space $\mathcal{S} = \{\mathbf{s} = (s_1, s_2, \cdots) : 1 \geq s_1 \geq s_2 \geq \cdots \geq 0 \text{ with } s_i = \frac{k_i}{N} \text{ for some integer } k_i\}$. By convention, we denote $s_0 = 1$ throughout the context.

We consider two types of performance measures: the response time and the average queue length. which we define as follows. 1) Let $c_w : \mathcal{S} \to \mathbb{R}^+$ denote the (random) response time (waiting time + service time) of a job arriving when the underlying system is in state $s$ just before the arrival. 2) For the average queue length, we define $c_q : \mathcal{S} \to \mathbb{R}^+$ as the corresponding cost function, where $c_q(s)$ represents the average queue length (include tasks in service) across all $N$ servers in state $s$.

Equivalently,

$$c_q(s) := \frac{1}{N} \sum_{i=1}^{N} q_i = \sum_{i=1}^{\infty} s_i.$$

Then, we consider the long-run average response time and long-run average queueing length, defined as

$$C_w(\lambda) := \lim_{T \to \infty} \frac{1}{TN\lambda} \sum_{t_j < T} c_w(s(t_j^-)), \quad C_q(\lambda) := \lim_{T \to \infty} \frac{1}{TN\lambda} \sum_{t_j < T} c_q(s(t_j^-)), \qquad (2)$$

where $s(t_j^-)$ denotes the state just before the arrival of $j$-th task, namely $s(t_j^-) = \lim_{t \uparrow t_j} s(t)$.

In general, our objective is to study the global treatment effect (GTE), defined as the performance difference between two scheduling policies in a parallel-server system. For clarity of exposition, we focus on the GTE between the power-of-$d_1$ policy and the power-of-$d_2$ policy, where $1 \le d_2 < d_1$. Recall that power-of-$d$ scheduling policy assigns each incoming task to the shortest queue among $d$ servers chosen uniformly at random. We remark that our methodology extends naturally to arbitrary scheduling policies, as we demonstrate in Section 5. We represent the treatment assignment by $a \in \{0, 1\}$, where $a = 0$ denotes the control policy and $a = 1$ denotes the treatment policy.

Therefore, with a slight abuse of notation, we define $c_q(s, a)$ and $c_w(s, a)$ to denote the queue length and response time metrics, respectively, under treatment assignment $a \in \{0, 1\}$. Furthermore, we use $C_{0,w}(\lambda)$ and $C_{1,w}(\lambda)$ to denote the performance measures under the global control and global treatment regimes. It is then customary to consider the estimation of the Global Treatment Effect (GTE), defined as

$$\text{GTE}_w = C_{1,w}(\lambda) - C_{0,w}(\lambda). \qquad (3)$$

As our main performance metric in this paper is the response time, we may sometimes omit the subscript $w$ for simplicity.

The following proposition presents Little's law, which is central to our estimator design. It states that the GTE defined in terms of response time and that defined in terms of queue length are equivalent up to a scaling factor given by the arrival rate.

**Proposition 1.** *We define the queue-length-based GTE as*

$$\text{GTE}_q = \frac{C_{1,q}(\lambda) - C_{0,q}(\lambda)}{\lambda}. \qquad (4)$$

*Then we have*

$$\text{GTE}_w = \text{GTE}_q. \qquad (5)$$

Proof of Proposition 1 is in Appendix B.

## 4 EXPERIMENTATION AND ESTIMATION

### 4.1 RANDOMIZATION AND THE NAIVE ESTIMATOR

In this paper, we focus on the commonly used Bernoulli experiments, conducted over the time interval $[0, T]$. Specifically, given $0 < p < 1$, we apply the control policy (power-of-$d_1$ policy) to each incoming task independently with probability $1 - p$, and apply the treatment policy (power-of-$d_2$ policy) with probability $p$. For convenience, we refer to this randomized assignment mechanism as the power-of-$d_1/d_2$ policy. Under this setup, the collected data consists of a sequence of vectors

$$\left\{ \left( t_j, a(t_j), \mathbf{l}(t_j^-, a(t_j)), \iota(t_j) \right) : j = 0, 1, \cdots \right\}. \qquad (6)$$

Here $t_j$ denotes the arrival time of the $j$-th task and $a(t_j)$ denotes the corresponding action. According to the definition of the power-of-$d$ policy, we are actually unable to fully observe the state $s(t_j^-)$ just before time $t_j$. Instead, we can only partially observe the queue lengths in those $d$ randomly chosen servers. Denote $\mathbf{l}(t_j^-, a(t_j)) = (l_1(t_j^-, a(t_j)), \ldots, l_d(t_j^-, a(t_j)))$ as the $d$-dimensional real-valued vector recording the observed queue lengths when the $j$-th task arrives, where $d = d_1$ if $a(t_j) = 0$ and $d = d_2$ otherwise. Furthermore, $\iota(t_j) \in \{1, 2, \cdots, N\}$ denotes the assignment of the task, i.e., $\iota(t_j) = i$ means the $j$-th task is assigned to the $i$-th server according to the chosen policy.

Following the definitions of costs in Section 3, we adopt the data-oriented estimations for costs

$$\hat{c}_{w,j} := \mu_{\iota(t_j)}^{-1}(l_{\iota(t_j)}(t_j^-, a(t_j)) + 1), \quad \hat{c}_{q,j} := \frac{1}{d} \sum_{k=1}^{d} l_k(t_j^-, a(t_j)). \tag{7}$$

Indeed, the estimated response-time-based cost is slightly different from the original response-time-based cost. The reasons that we make this adjustment are as follows. Firstly, the true response time for a task is not observable at its arrival time. Only until the task leaves the system can we document its exact response time. This leads to computational inconvenience and a possible non-Markov property. In contrast, the estimated response-time-based cost can be observed and documented as long as we have an in-hand estimation of the service rates. Secondly, the service rates $\mu$ are known or easy to estimate in practice and the estimated cost (7) also contributes to variance reduction.

For ease of exposition, we denote by $\mathcal{C} = \{j : t_j < T, \ a(t_j) = 0\}$ and $\mathcal{T} = \{j : t_j < T, \ a(t_j) = 1\}$ the sets of tasks that are assigned by the control policy and the treatment policy, respectively. Furthermore, we denote $n_C = |\mathcal{C}|$ and $n_T = |\mathcal{T}|$ the number of tasks that are assigned by the two policies. The naive estimator for the GTE is given by

$$\widehat{\text{GTE}}_{\text{NV}} = \frac{1}{n_T} \sum_{j \in \mathcal{T}} \hat{c}_{w,j} - \frac{1}{n_C} \sum_{j \in \mathcal{C}} \hat{c}_{w,j}. \tag{8}$$

Heuristically, the naive estimator measures the empirical difference in average response times between the control and treatment policies, based on sample paths generated under the mixing policy. Since $d_2 < d_1$, the GTE is positive and the naive estimator ignores interference effects, which leads to a systematic underestimation. In particular, when estimating the average response time under the power-of-$d_2$ policy using $\frac{1}{n_T} \sum_{j \in \mathcal{T}} \hat{c}_{w,j}$, the estimator underestimates the true value because queue lengths are shorter than they would be under the global treatment regime, owing to the presence of the power-of-$d_1$ policy. Conversely, the average response time under the power-of-$d_1$ policy is overestimated. This systematic bias highlights the importance of developing alternative estimators with improved accuracy.

### 4.2 DIFFERENCES-IN-Q ESTIMATORS

Our next goal is to tackle the Markovian interference and propose better estimations for GTE defined in (3). Following Farias et al. (2022), we consider the non-discounted $Q$ function

$$Q(s(t_j^-), a(t_j)) := \sum_{k=j}^{\infty} \mathbb{E}\left[\left(c(s(t_k^-), a(t_k)) - \bar{c}\right)\right], \tag{9}$$

where $c(\cdot, \cdot)$ represents $c_w(\cdot, \cdot)$ or $c_q(\cdot, \cdot)$ and $\bar{c}$ is the long run average cost under the mixing policy. Due to the Markov property of the system, the Q-function in (9) is well-defined in the sense that $Q(s(t_j^-), a(t_j)) = Q(s(t_j'^-), a(t_j'))$ if $s(t_j^-) = s(t_j'^-)$ and $a(t_j) = a(t_j')$. The finiteness of $Q(\cdot, \cdot)$ is due to the rapid mixing property of the underlying system (Luczak and McDiarmid, 2006, Theorem1.1). Then, Farias et al. (2022) propose using

$$\widehat{\text{GTE}}_{\text{DQ}} := \frac{1}{n_T} \sum_{j \in \mathcal{T}} Q(s(t_j^-), 1) - \frac{1}{n_C} \sum_{j \in \mathcal{C}} Q(s(t_j^-), 0) \tag{10}$$

to estimate the global treatment effect (GTE). Farias et al. (2022) further show that $\widehat{\text{GTE}}_{\text{DQ}}$ can significantly reduce the bias compared to the naive estimator, $\widehat{\text{GTE}}_{\text{NV}}$. Intuitively, the DQ estimators account not only for the immediate cost difference between the two policies, but also for all future downstream effects caused by the different state transitions that each policy induces. In other words, they capture both the direct reward difference and the subsequent impact of state evolution.

To estimate $Q(s(t_j^-), a(t_j))$, we adopt the simplest Monte Carlo estimation. The response-time Q and Queue-length Q functions are defined as

$$\hat{Q}_{w,j}^L = \sum_{k=0}^{L} (\hat{c}_{w,j+k} - \bar{c}), \quad \hat{Q}_{q,j}^L = \sum_{k=0}^{L} (\hat{c}_{q,j+k} - \bar{c}). \tag{11}$$

Here, $L$ serves as a hyperparameter that truncates the infinite sum in the original definition of the Q-functions. We then propose response-time-based and queue-length-based Differences-in-Q (DQ) estimators to estimate the GTE:

$$\widehat{DQ}_w^L = \frac{1}{n_T}\sum_{j\in\mathcal{T}}\hat{Q}_{w,j}^L - \frac{1}{n_C}\sum_{j\in\mathcal{C}}\hat{Q}_{w,j}^L = \frac{1}{n_T}\sum_{j\in\mathcal{T}}\sum_{k=0}^L \hat{c}_{w,j+k} - \frac{1}{n_C}\sum_{j\in\mathcal{C}}\sum_{k=0}^L \hat{c}_{w,j+k}, \tag{12}$$

$$\widehat{DQ}_q^L = \frac{1}{n_T\lambda}\sum_{j\in\mathcal{T}}\hat{Q}_{q,j}^L - \frac{1}{n_C\lambda}\sum_{j\in\mathcal{C}}\hat{Q}_{q,j}^L = \frac{1}{n_T\lambda}\sum_{j\in\mathcal{T}}\sum_{k=0}^L \hat{c}_{q,j+k} - \frac{1}{n_C\lambda}\sum_{j\in\mathcal{C}}\sum_{k=0}^L \hat{c}_{q,j+k}. \tag{13}$$

Here, the normalization term $\frac{1}{\lambda}$ in (13) comes from Little's Law, as introduced in Appendix A and it is easy to estimate as detailed in Appendix C.2. Note that we do not need to estimate $\bar{c}$, as it cancels out in the treatment effect estimation. Then, Little's law further indicates that $\widehat{DQ}_w^L$ and $\widehat{DQ}_q^L$ should have similar expectations.

### 4.3 MIXED DIFFERENCES-IN-Q ESTIMATOR

Although the DQ estimators proposed in Section 4.2 greatly reduce bias, both types of the Differences-in-Q estimators suffer from the large variances accumulated by the sum of $L+1$ subsequent costs. For instance, numerical results from Table 1 suggest that the standard deviations of queue-length-based and response-time-based Differences-in-Q estimators are generally 50-500 times larger than the standard deviation of the naive estimator. Therefore, it is necessary to investigate a method to reduce variance.

Inspired by Little's Law, we combine queue-length-based Q-functions with response-time-based Q-functions to address the high variance problem. Empirically, we observe that the queue-length-based Q-functions and the response-time-based Q-functions are strongly correlated. Hence, we apply 'control variate' technique to reduce variance (Ross, 2022). Specifically, we define mixed Q-functions as follows:

$$\hat{Q}_{\text{mix}}^{\alpha,L} = \alpha\hat{Q}_w^L + (1-\alpha)\frac{\hat{Q}_q^L}{\lambda}, \tag{14}$$

where $\alpha$ is a weight that we will optimize to minimize the variance. The variance of the mixed Q-function can be explicitly calculated as a quadratic polynomial of $\alpha$:

$$\begin{aligned}\text{Var}(\hat{Q}_{\text{mix}}^{\alpha,L}) =& \alpha^2\text{Var}(\hat{Q}_w^L) + \frac{(\alpha-1)^2}{\lambda^2}\text{Var}(\hat{Q}_q^L) - \frac{2\alpha(\alpha-1)}{\lambda}\text{Cov}(\hat{Q}_w^L,\hat{Q}_q^L)\\ =& \alpha^2\left(\text{Var}(\hat{Q}_w^L) + \frac{\text{Var}(\hat{Q}_q^L)}{\lambda^2} - \frac{2\text{Cov}(\hat{Q}_w^L,\hat{Q}_q^L)}{\lambda}\right)\\ & + 2\alpha\left(\frac{\text{Cov}(\hat{Q}_w^L,\hat{Q}_q^L)}{\lambda} - \frac{\text{Var}(\hat{Q}_q^L)}{\lambda^2}\right) + \frac{\text{Var}(\hat{Q}_q^L)}{\lambda^2}.\end{aligned} \tag{15}$$

This representation leads to an optimal choice of $\alpha$, i.e.,

$$\alpha^* := \operatorname*{argmin}_\alpha\{\text{Var}(\hat{Q}_{\text{mix}}^{\alpha,L})\} = \frac{\text{Var}(\hat{Q}_q^L) - \lambda\text{Cov}(\hat{Q}_w^L,\hat{Q}_q^L)}{\lambda^2\text{Var}(\hat{Q}_w^L) + \text{Var}(\hat{Q}_q^L) - 2\lambda\text{Cov}(\hat{Q}_w^L,\hat{Q}_q^L)}. \tag{16}$$

To estimate $\alpha^*$, we estimate the variance and covariance in (16) using their sample version.

Substituting the estimated optimal value of $\alpha$ (16) and the truncated Q-functions (11) into (14) yields the estimated optimal mixed Q-functions for the $j$-th task

$$\hat{Q}_{\text{mix},j}^{\hat{\alpha},L} = \hat{\alpha}\hat{Q}_{w,j}^L - \frac{\hat{\alpha}-1}{\lambda}\hat{Q}_{q,j}^L. \tag{17}$$

Hence, we obtain the optimal mixed Differences-in-Q estimator in the pre-limiting system

$$\widehat{DQ}_{\text{mix}}^L = \frac{1}{n_T}\sum_{j\in\mathcal{T}}\hat{Q}_{\text{mix},j}^{\hat{\alpha},L} - \frac{1}{n_C}\sum_{j\in\mathcal{C}}\hat{Q}_{\text{mix},j}^{\hat{\alpha},L}. \tag{18}$$

It is easy to see that $\widehat{DQ}_{\text{mix}}^L$ retains the bias-reduction properties of both the response-time-based and queue-length-based DQ estimators.

## 5 NUMERICAL RESULTS

We conduct numerical experiments to show the efficacy of Differences-in-Q estimators in estimating GTE. Unless otherwise declared, we choose $N = 20$ and $p = 0.5$ throughout the following experiments. The total simulation time in each experiment is chosen to be $10^6$, which also means there are approximately $NT\lambda = 2\lambda \times 10^7$ jobs arriving to the many-server system during each experiment. The truncation length is set as $L = \lfloor 30N\lambda \rfloor$. We run 100 independent replications to evaluate the bias and variance of different estimators.

We summarize homogeneous service time results in Section 5.1 and leave heterogeneous service time results, as well as the formulation of the group estimator, in Section 5.2. Section 5.3 includes non-stationary arrival rate experiments. Section 5.4 conducts the experiments for the constant service time. Finally, Section 5.5 presents an ablation study on different truncation lengths and shows that our results are not sensitive to this choice. We primarily focus on the power-of-$d_1/d_2$ experiments with $d_1 = 3$ and $d_2 = 2$. Additional results from the power-of-5/3 experiments are presented in D.1. Inspired by Farias et al. (2023), we also propose the doubly robust Differences-in-Q estimators in Section D.2 and compare their efficiency with the Differences-in-Q estimators. Section D.3 includes experiments concerning other well-known load balancing policies. Finally, D.4 induces experiments with communication delay.

### 5.1 HOMOGENEOUS SERVICE TIME

We present a direct comparison of several estimators in Table 1. These include the naive estimator, queue-length-based, response-time-based and mixed DQ estimator. To facilitate a clear understanding of our results, we present violin curves for experiments with $\lambda \in \{0.7, 0.8, 0.85, 0.9, 0.95\}$ in Figures 2(a)-2(e). Moreover, we draw the mean squared error (MSE) graphically in Figure 2(f).

For ease of exposition, we abbreviate 'estimator' as 'Est.' and 'standard deviation' as 'Std. Dev.' in table indices, where the estimator is reported as the sample average of 100 independent replications. As can be seen from Table 1, the naive estimator demonstrates the largest bias yet the smallest variance among the estimators. The queue-length-based and response-time-based DQ estimators exhibit reductions in bias but at the expense of increased variances. Moreover, the mixed DQ estimator notably reduces variance through the utilization of the 'control variate' technique. Figure 2 further confirms that the mean squared error (MSE) of our mixed DQ estimator is the lowest among all estimators across various arrival rates. The mixed DQ estimator therefore turns out to be a powerful tool to estimate the GTE in a real-world queue system.

Table 1: Power-of-$d_1/d_2$ experiments with $d_1 = 3$, $d_2 = 2$ in the homogeneous service time setting.

| $\lambda$ | Index | GTE | Naive | qDQ | wDQ | mixDQ |
|---|---|---|---|---|---|---|
| $\lambda = 0.8$ | Est. | 0.358 | 0.254 | 0.348 | 0.348 | 0.348 |
| | Std. Dev. | | $4.92 \times 10^{-4}$ | 0.082 | 0.053 | 0.016 |
| $\lambda = 0.85$ | Est. | 0.439 | 0.283 | 0.407 | 0.414 | **0.430** |
| | Std. Dev. | | $5.37 \times 10^{-4}$ | 0.097 | 0.070 | 0.026 |
| $\lambda = 0.9$ | Est. | 0.566 | 0.316 | 0.465 | 0.481 | **0.540** |
| | Std. Dev. | | $5.00 \times 10^{-4}$ | 0.152 | 0.121 | 0.042 |
| $\lambda = 0.95$ | Est. | 0.797 | 0.360 | 0.435 | 0.471 | **0.737** |
| | Std. Dev. | | $8.19 \times 10^{-4}$ | 0.351 | 0.315 | 0.115 |

In the 'Index' column, we abbreviate 'Estimator' as 'Est.' and 'Standard Deviation' as 'Std. Dev.'. In the first row, 'qDQ', 'wDQ' and 'mixDQ' refer to the queue-length-based, the response-time-based and the mixed DQ estimators, respectively, as defined in (13) and (18).

In Table 1 and Figure 2, we observe that for small values of $\lambda$, all DQ estimators demonstrate effective bias reduction. For relatively large $\lambda$, the mixed DQ estimator not only reduces variance but also, unexpectedly, minimizes bias. This behavior is attributed to the truncation parameter $L$, which introduces some truncation bias. Although Little's Law suggests that the response-time-based, queue-

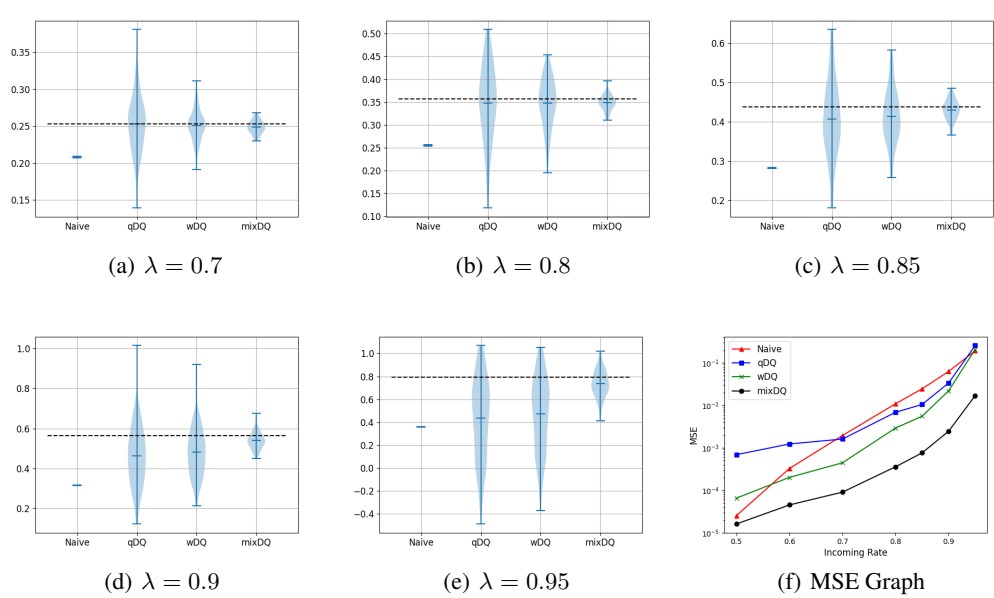

Figure 2: Power-of-3/2 experiments in the homogeneous service time setting.

length-based, and mixed DQ estimators exhibit similar biases, empirical results indicate that the response-time-based estimator converges faster than the queue-length-based estimator as $L \to \infty$. Furthermore, we observe that $\hat{\alpha} > 1$. Consequently, the mixed DQ estimator exhibits the fastest convergence and the smallest bias among all three DQ estimator types.

## 5.2 HETEROGENEOUS SERVICE TIME

In this subsection, we conduct numerical experiments to evaluate our estimators under conditions of heterogeneous service rates. Throughout this subsection, we set the number of servers to $N = 20$ and the service rates to range from $0.90$ to $1.09$ in increments of $0.01$. Prior to presenting numerical data, we introduce the group estimator as follows.

When constructing the group estimator, we initially partition all the servers randomly into two equal-sized groups, labeled as Group 1 and Group 2 (assuming $N$ is even). This partitioning is performed at the beginning of each experiment and remains consistent throughout the replication. Upon the arrival of a new job, it will be assigned to Group 2 using the power-of-$d_2$ policy with probability $p$ and to Group 1 using the power-of-$d_1$ policy otherwise. Average response times are then calculated separately for each group, yielding the group estimator:

$$\widehat{\text{GTE}}_{\text{group}} = \frac{1}{n_T} \sum_{j \in \mathcal{T}} \hat{c}_{w,j} - \frac{1}{n_C} \sum_{j \in \mathcal{C}} \hat{c}_{w,j}, \qquad (19)$$

It's important to highlight that the group experiments do not suffer from the previously discussed Markovian interference, as distinct policies are applied to separate sets of servers. Our simulation results, outlined in Table 2 and plotted in Figure 3, encompass the naive estimator, the group estimator, and three types of the DQ estimators.

Consistent with other experiments, the naive estimator shows the highest bias but the lowest variance among the estimators. The queue-length-based and response-time-based DQ estimators demonstrate reduced bias, albeit with increased variance. Moreover, the mixed DQ estimator notably reduces variance and bias. Although the group estimator displays a comparatively modest bias compared to the naive estimator, it manifests the highest variance among all the estimators. It's also worth noting that prolonging the simulation time won't alleviate the variance of the group estimator since the separation of the two groups occurs independently and randomly at the beginning of each replication, introducing a significant but non-vanishing variance. Overall, the mixed DQ estimator maintains its effectiveness even under conditions of heterogeneous service rates.

Table 2: Power-of-3/2 experiments in the heterogeneous service time setting.

| $\lambda$ | Index | GTE | Naive | Group | qDQ | wDQ | mixDQ |
|---|---|---|---|---|---|---|---|
| $\lambda = 0.85$ | Est. | 0.458 | 0.291 | 0.468 | 0.446 | 0.446 | **0.448** |
| | Std. Dev. | | $5.46 \times 10^{-4}$ | 0.055 | 0.099 | 0.073 | 0.027 |
| $\lambda = 0.9$ | Est. | 0.595 | 0.327 | 0.638 | 0.509 | 0.523 | **0.573** |
| | Std. Dev. | | $6.35 \times 10^{-4}$ | 0.114 | 0.172 | 0.140 | 0.051 |
| $\lambda = 0.95$ | Est. | 0.859 | 0.374 | 1.019 | 0.437 | 0.478 | **0.808** |
| | Std. Dev. | | $8.61 \times 10^{-4}$ | 0.383 | 0.343 | 0.312 | 0.138 |

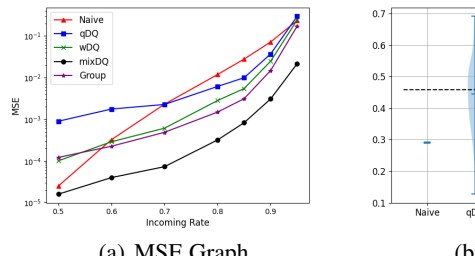 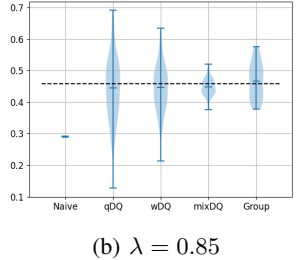 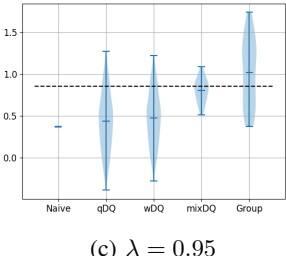

(a) MSE Graph      (b) $\lambda = 0.85$      (c) $\lambda = 0.95$

Figure 3: Power-of-3/2 experiments in the heterogeneous service time setting.

## 5.3 NON-STATIONARY ARRIVAL RATE RESULTS

We implement non-stationary arrival rate experiments and compare the efficiency of both naive estimator and DQ estimators. We set the arrival rate to vary in a cyclic way, i.e., $\lambda(t) = 0.9 + 0.15\sin(t)$. Numerical data are documented and exhibited in Table 3 and Figure 4 below. Similar to other experimental results, our proposed mixed DQ estimators consistently outperform other estimators, exhibiting low bias and variance. Additionally, they achieve the lowest MSE among all estimators.

Table 3: Power-of-3/2 experiments with non-stationary arrivals.

| $\lambda$ | Index | GTE | Naive | qDQ | wDQ | mixDQ |
|---|---|---|---|---|---|---|
| $\lambda(t) = 0.9 + 0.15\sin(t)$ | Est. | 0.561 | 0.316 | 0.465 | 0.482 | **0.547** |
| | Std. Dev. | | $6.04 \times 10^{-4}$ | 0.134 | 0.108 | 0.044 |
| | MSE | | 0.060 | 0.027 | 0.018 | **0.002** |

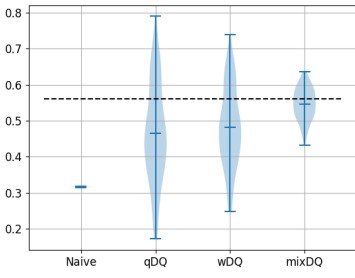

Figure 4: Power-of-$d_1/d_2$ Experiments with $d_1 = 3$, $d_2 = 2$ and non-stationary arrivals.

## 5.4 GENERAL SERVICE TIME EXPERIMENTS

In this section, we include the following experiment where the service time is constant and equal to the inverse of the service rate. This serves as a representative example of a general (non-exponential) service-time distribution.

Table 4: Power-of-$d_1/d_2$ experiments with $d_1 = 3$, $d_2 = 2$ in the constant-service-time setting.

| $\lambda$ | Index | GTE | Naive | qDQ | wDQ | mixDQ |
|---|---|---|---|---|---|---|
| $\lambda = 0.85$ | Est. | 0.201 | 0.132 | 0.191 | 0.193 | **0.196** |
| | Std. Dev. | | $2.78 \times 10^{-4}$ | 0.041 | 0.026 | 0.009 |
| $\lambda = 0.9$ | Est. | 0.261 | 0.149 | 0.243 | 0.245 | **0.251** |
| | Std. Dev. | | $2.87 \times 10^{-4}$ | 0.070 | 0.051 | 0.017 |
| $\lambda = 0.95$ | Est. | 0.375 | 0.170 | 0.278 | 0.288 | **0.348** |
| | Std. Dev. | | $4.10 \times 10^{-4}$ | 0.145 | 0.127 | 0.052 |

In the 'Index' column, 'Est.' abbreviates 'Estimator' and 'Std. Dev.' abbreviates 'Standard Deviation'. The labels 'qDQ', 'wDQ', and 'mixDQ' refer to the queue-length-based, response-time-based, and mixed DQ estimators, respectively.

From the experimental results, we observe the Differences-in-Q estimators have smaller bias compared to the naive estimator. Furthermore, the mixed Differences-in-Q estimator show significantly smaller variance compared to the other counterparts. This further validates the effectiveness of our method in a general service time setting.

### 5.5 ABLATIONS ON THE TRUNCATION LENGTH

In this section, we consider the ablation study on the truncation length $L$. We take the power-of-$3/2$ experiments with $\lambda = 0.9$ and $N = 20$ as an example. We consider four different truncation lengths, i.e., $L = 10N\lambda$, $L = 30N\lambda$, $L = 60N\lambda$ and $L = 100N\lambda$. Experimental results are shown below.

Table 5: Power-of-$3/2$ Experiments with $N = 20$ and varying $L$. GTE is 0.566. All data are shown in Est. (Std. Dev.) format. Standard deviation of the naive estimator is omitted, as it is extremely small.

| $\lambda = 0.9$ | Naive | qDQ | wDQ | mixDQ |
|---|---|---|---|---|
| $L = 10N\lambda$ | 0.316 | 0.391(0.039) | 0.404(0.030) | 0.537(0.012) |
| $L = 30N\lambda$ | 0.316 | 0.462(0.067) | 0.479(0.054) | 0.544(0.021) |
| $L = 60N\lambda$ | 0.316 | 0.533(0.118) | 0.537(0.097) | 0.550(0.033) |
| $L = 100N\lambda$ | 0.316 | 0.557(0.139) | 0.556(0.114) | 0.551(0.046) |

Under the same experimental settings, we observe the bias-variance trade-off. When the truncation length $L$ is small, the variance is typically lower, but the Q-functions suffer from bias due to insufficient trajectory length. Conversely, when $L$ is large, the Q-function estimates converge better, but the variance increases significantly, which limits the estimator performance. Based on this observation, we select $L = 30N\lambda$ to balance the bias and variance. Additionally, we find that the mixed Differences-in-Q estimator converges faster than its counterparts when $L$ is small. This further demonstrates the effectiveness of our proposed method.

## 6 CONCLUSION

In this paper, we present an A/B testing framework for load balancing policies and propose a mixed Differences-in-Q estimator to significantly reduce bias and variance in estimating the Global Treatment Effect (GTE). For future work, we aim to rigorously develop inference procedures for our estimator and demonstrate its robustness in more complex data center scenarios, including large language model (LLM) inference.

More broadly, we plan to extend this idea of leveraging domain knowledge to improve evaluation methodologies. For example, in assessing demand-prediction models for inventory control, we aim to exploit the well-known optimality of the $s, S$ policy and develop a decision-focused scoring rule—grounded in the newsvendor-type loss—to more accurately evaluate prediction quality in terms of operational performance. Furthermore, for applications such as dynamic pricing, we plan to exploit structural properties—such as revenue concavity, value-function convexity, and price monotonicity—to design decision-focused evaluation metrics that more directly capture how prediction errors affect pricing performance.

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

## APPENDIX A  LITTLE'S LAW

In this section, we briefly introduce and summarize Little's law in queueing theory. Little's law, first proposed in Cobham (1954); Morse (2004) and proved in Little (1961); Whitt (1991); Kim and Whitt (2013), has been studied under different situations for various purposes and in a wide range of fields. In general, Little's Law studies a parallel-server system over finite time period. Suppose the queue length in the current parallel-server system at time t is $l(t)$. The average queue length over time period $[0, T]$, L, is defined by

$$L = \frac{1}{T} \int_0^T l(t)dt. \tag{20}$$

Suppose further that there are $S(T)$ tasks arriving at the parallel-server system during the period $(0, T]$. The average arrival rate, $\lambda$, is defined by

$$\lambda = \frac{S(T)}{T}. \tag{21}$$

For each task, the response time is defined as the gap between its arrival time and departure time. Denote $W_i$ the response time of the i-th task. The average response time among these $S(T)$ tasks is defined by

$$W = \frac{1}{S(T)} \sum_{i=1}^{S(T)} W_i. \tag{22}$$

Little's Law, however, states that the average queue length equals to the average arrival rate multiplied by the average response time, i.e.,

$$L = \lambda W. \tag{23}$$

Initially, Little's Law was studied in a parallel-server system with both an empty state at $t = 0$ and an empty state at $t = T$. One exemplary application of this system is a supermarket, which opens empty in the morning and closes empty in the evening. A heuristic proof is as follows. Suppose that the arrival time and departure time of the i-th task are $t_{1,i}$ and $t_{2,i}$. Then

$$W_i = t_{2,i} - t_{1,i} = \int_0^T \mathbf{1}_{\{t_{1,i} < t < t_{2,i}\}} dt. \tag{24}$$

This further leads to

$$\lambda W = \frac{1}{T} \sum_{i=1}^{S(T)} \int_0^T \mathbf{1}_{\{t_{1,i} < t < t_{2,i}\}} dt = \frac{1}{T} \int_0^T \sum_{i=1}^{S(T)} \mathbf{1}_{\{t_{1,i} < t < t_{2,i}\}} dt = \frac{1}{T} \int_0^T l(t)dt = L. \tag{25}$$

Later, Little's Law was generalized to permit non-zero starting and ending queue lengths. The generalization was constructed at the expense of redefining the average arrival rate to include the tasks that are already in the system at $t = 0$. This leads to the general version of Little's Law, as stated in equation (23). For a more detailed description and overview of the theory and applications of Little's Law, one may refer to Little (2011); Wolff (2011).

## APPENDIX B PROOF OF PROPOSITION 1

**Proof of Proposition 1.** In order to prove that the response-time-based and queue-length-based GTE are the same, it suffices to show

$$C_{i,q}(\lambda) = \lambda C_{i,w}(\lambda) \text{ for } i = 0, 1. \tag{26}$$

We first consider the $i = 0$ case. According to Mitzenmacher (2001) and Luczak and Mc-Diarmid (2006), the marginal distribution of the process will converge to the stationary distribution as $T \to \infty$, due to the ergodicity. Therefore, we have $C_{0,q}(\lambda) = \mathbb{E}_{s \sim \pi_0}[c_q(s)]$ and $C_{0,w}(\lambda) = \mathbb{E}_{s \sim \pi_0}[c_w(s)]$, where $\pi_0$ is the stationary distribution under the control policy. Now consider a parallel queue service system that has initial distribution $\pi_0$ and is implemented with the control policy. For a given period $[0, T]$, we denote $L_T, W_T, \lambda_T$ the average queue length, average response time and average arrival rate, respectively. By Little's law, we have

$$L_T = \lambda_T W_T. \tag{27}$$

In addition, by the definition of stationary distribution, we know all marginal distributions of this system is $\pi_0$. Hence, we have $\mathbb{E}[L_T] = \mathbb{E}_{s \sim \pi_0}[N \sum s_i] = N\mathbb{E}_{s \sim \pi_0}[c_q(s)]$ and $\mathbb{E}[W_T] = \mathbb{E}_{s \sim \pi_0}[c_w(s)]$. By taking expectation on both sides of (27), we obtain

$$N\mathbb{E}_{s \sim \pi_0}[c_q(s)] = N\lambda\mathbb{E}_{s \sim \pi_0}[c_w(s)]. \tag{28}$$

Here, we use the fact that $\lambda_T \xrightarrow{a.s.} N\lambda$ as $T \to \infty$. This leads to $C_{0,q}(\lambda) = \lambda C_{0,w}(\lambda)$. Similarly, we also have $C_{1,q}(\lambda) = \lambda C_{1,w}(\lambda)$, which yields the proposition.

## APPENDIX C IMPLEMENTATION

### C.1 OBSERVABLE QUEUEING INFORMATION FOR OTHER SCHEDULING POLICIES

Following Section 4.1, we illustrate the observable queueing information for both the MJSQ-$r$ policy and the JIQ-$d$ policy. Suppose that the j-th task is chosen to be assigned by the MJSQ-$r$ policy. At this point, we can still observe the vector

$$\{(t_j, a(t_j), \mathbf{l}(t_j, a(t_j)), \iota(t_j))) : \ j = 0, 1, \cdots \}, \tag{29}$$

where $\mathbf{l}(t_j, a(t_j))$ is the single queue length of the server if the task is randomly assigned to one of the server and $\mathbf{l}(t_j, a(t_j))$ is a $N$-dimensional vector recording the queue lengths from all the servers if the task is assigned by the JSQ policy.

On the other hand, when the $j$-th task is assigned by the JIQ-$d$ policy, the observed vector can be recorded with the same form to (29). In this case, the vector $\mathbf{l}(t_j, a(t_j))$ is a $d$-dimensional vector if there's no empty server. When there's at least one of the server from the system that sends the empty signal to the central dispatcher, we randomly draw a server from the system and record its queue length in $\mathbf{l}(t_j, a(t_j))$.

### C.2 DIFFERENCES-IN-Q CALCULATION

We illustrate our implementation of the Differences-in-Q estimators and the mixed Differences-in-Q estimators proposed in Section 4. To address the non-stationary arrival rate experiments, we start by introducing the empirical estimation for the incoming rate $\lambda$. We define

$$\hat{\lambda} = \frac{n_C + n_T}{T}. \tag{30}$$

Clearly, $\hat{\lambda}$ is a consistent estimator for the incoming rate $\lambda$. Next, we recall that the response-time-based cost defined in (7) relies directly on the service rates $\mu_1, \cdots, \mu_N$. If the service rates are not a priori known, we provide an estimation method below. Let $A_i = \{t_j < T : \iota(t_j) = i\}$ be the set of arrival times in the i-th server. We define

$$\hat{\mu}_i = \frac{\sum_{t_j \in A_i} \min_{1 \leq k \leq d}(l_k(t_j, a(t_j)) + 1)}{\sum_{t_j \in A_i} w(t_j, a(t_j))}, \ \ 1 \leq i \leq N, \tag{31}$$

where $w(t_j, a(t_j))$ is the response time of the $j$-th job with treatment assignment $a(t_j)$. By the Strong Law of Large Numbers, we obtain

$$\hat{\mu}_i \overset{a.s.}{\to} \mu_i \quad as \ T \to \infty, \tag{32}$$

meaning that these estimations for the service rates are all consistent estimations. We then estimate the response-time-based cost by

$$\hat{c}_{w,j} = \hat{\mu}_{n(t_j)}^{-1} \min_{1 \leq k \leq d} (l_k(t_j, a(t_j)) + 1). \tag{33}$$

### C.3 Inference procedure and variance estimation

In this subsection, we give a method to estimate variance and conduct inference. we adopt the following simpler estimations of the standard errors:

$$\widehat{\mathrm{SE}}_w := 2\sqrt{\frac{\mathrm{Var}(\hat{Q}_w)}{TN\hat{\lambda}}}, \tag{34}$$

$$\widehat{\mathrm{SE}}_q := \frac{2}{\hat{\lambda}}\sqrt{\frac{\mathrm{Var}(\hat{Q}_q)}{TN\hat{\lambda}}}, \tag{35}$$

$$\widehat{\mathrm{SE}}_{\mathrm{mix}} := 2\sqrt{\frac{\mathrm{Var}(\hat{Q}_{\mathrm{mix}})}{TN\hat{\lambda}}}. \tag{36}$$

On the other hand, we also calculate the standard error for the naive estimator and the group estimator (see Section 5.2 for its construction). Since both of the two estimators don't involve Q-functions, we follow the original definition to calculate

$$\widehat{\mathrm{SE}} := \sqrt{\frac{\mathrm{Var}(\hat{c}_w(s,0))}{n_C} + \frac{\mathrm{Var}(\hat{c}_w(s,1))}{n_T}}. \tag{37}$$

We report the estimated standard error in Section D and compare it to the standard deviation of the corresponding estimators to show their consistency. Furthermore, we use the estimated standard error to construct asymptotic confidence intervals regarding the Differences-in-Q estimators in the following way. Given confidence rate $100(1-\delta)\%$, we choose $z > 0$ such that $\mathbb{P}(-z \leq \mathcal{N}(0,1) \leq z) = 1 - \delta$. We initiate

$$\hat{L}_N = \widehat{DQ}^L - z\widehat{\mathrm{SE}}, \quad \hat{R}_N = \widehat{DQ}^L + z\widehat{\mathrm{SE}}, \tag{38}$$

where $\widehat{DQ}^L$ is the value of the Differences-in-Q estimator and $\hat{SE}$ is calculated via (34). The interval $[\hat{L}_N, \hat{R}_N]$ serves as an asymptotic confidence interval with confidence rate $100(1-\delta)\%$ for the Differences-in-Q estimator.

## Appendix D  Additional Numerical Results

### D.1 Power-of-5/3 experiments in the homogeneous service time setting

In this subsection, we replicate the experiments in Section 5.1 but for power-of-5/3 experiments. We report detailed results for $\lambda \in \{0.5, 0.6, 0.7, 0.8, 0.85, 0.9, 0.95\}$ in Table 6 and Figure 5.

### D.2 Doubly Robust Differences-in-Q Estimator

Inspired by Farias et al. (2023); Jiang and Li (2016), we propose three types of doubly robust Differences-in-Q estimators and study the effects of variance reduction led by parametric estimations of Q-functions. Specifically, we define doubly robust Q-functions by

$$Q_\cdot^{\mathrm{DR}}(s,a) = Q_\cdot^{\mathrm{REG}}(s,a) + \frac{1_{\{p=a\}}}{p(a|s)}(Q_\cdot(s,a) - Q_\cdot^{\mathrm{REG}}(s,a)), \tag{39}$$

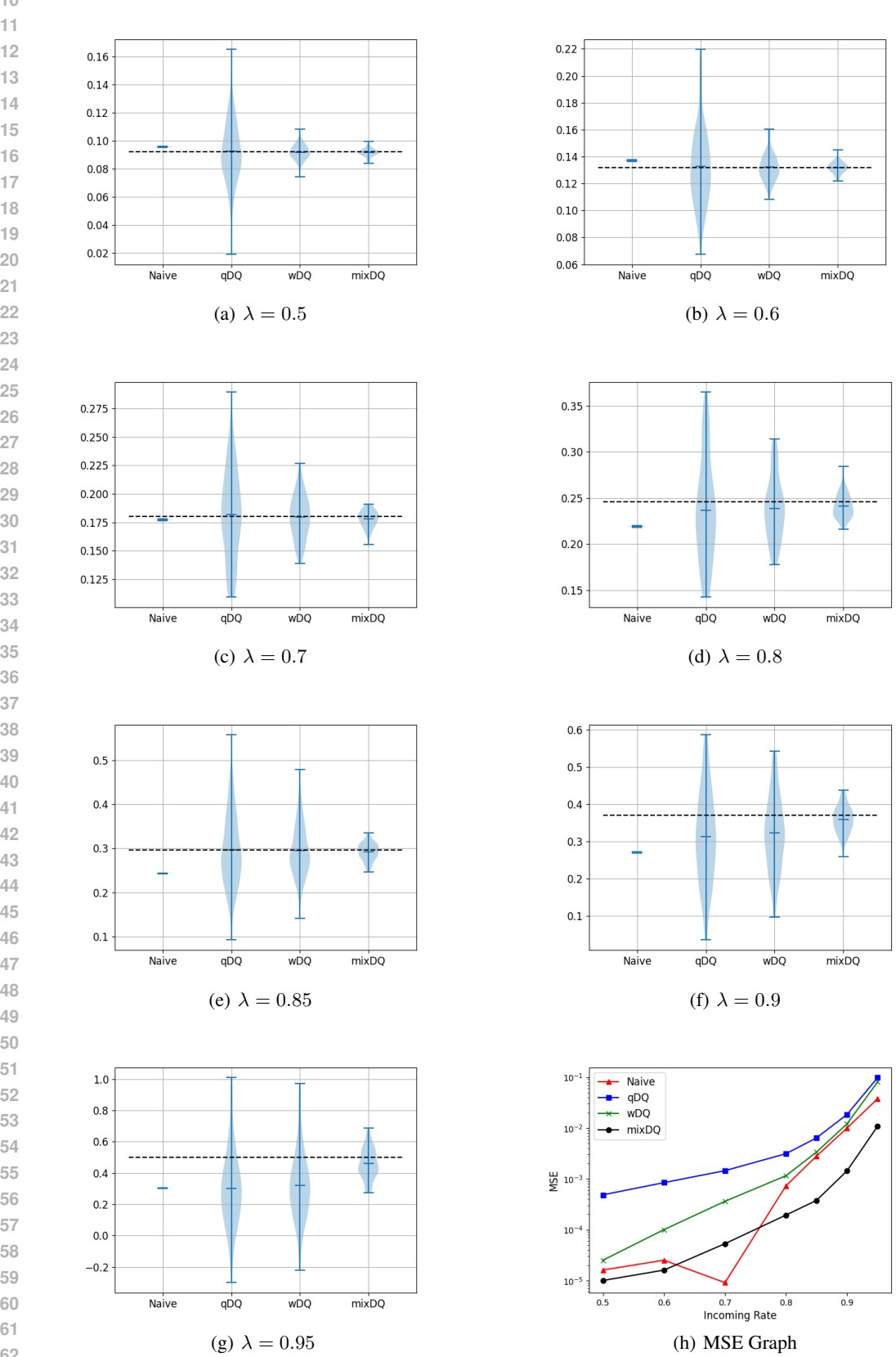

(a) $\lambda = 0.5$

(b) $\lambda = 0.6$

(c) $\lambda = 0.7$

(d) $\lambda = 0.8$

(e) $\lambda = 0.85$

(f) $\lambda = 0.9$

(g) $\lambda = 0.95$

(h) MSE Graph

Figure 5: Power-of-5/3 experiments in the homogeneous service time setting.

Table 6: Power-of-5/3 experiments in the homogeneous service time setting.

| $\lambda$ | Index | GTE | Naive | qDQ | wDQ | mixDQ |
|---|---|---|---|---|---|---|
| | Est. | 0.092 | 0.096 | 0.092 | 0.092 | 0.091 |
| $\lambda = 0.5$ | Std. Dev. | | $2.21 \times 10^{-4}$ | 0.022 | 0.005 | 0.003 |
| | Std. Err. | | $1.80 \times 10^{-4}$ | 0.023 | 0.006 | 0.003 |
| | Est. | 0.132 | 0.137 | 0.133 | 0.132 | 0.132 |
| $\lambda = 0.6$ | Std. Dev. | | $2.69 \times 10^{-4}$ | 0.029 | 0.010 | 0.004 |
| | Std. Err. | | $2.16 \times 10^{-4}$ | 0.029 | 0.010 | 0.004 |
| | Est. | 0.180 | 0.177 | 0.182 | 0.180 | 0.178 |
| $\lambda = 0.7$ | Std. Dev. | | $2.66 \times 10^{-4}$ | 0.038 | 0.019 | 0.007 |
| | Std. Err. | | $2.53 \times 10^{-4}$ | 0.038 | 0.018 | 0.007 |
| | Est. | 0.246 | 0.219 | 0.236 | 0.238 | 0.241 |
| $\lambda = 0.8$ | Std. Dev. | | $4.02 \times 10^{-4}$ | 0.055 | 0.033 | 0.013 |
| | Std. Err. | | $3.03 \times 10^{-4}$ | 0.060 | 0.037 | 0.013 |
| | Est. | 0.296 | 0.243 | 0.296 | 0.295 | 0.292 |
| $\lambda = 0.85$ | Std. Dev. | | $3.97 \times 10^{-4}$ | 0.080 | 0.058 | 0.019 |
| | Std. Err. | | $3.41 \times 10^{-4}$ | 0.082 | 0.058 | 0.020 |
| | Est. | 0.371 | 0.271 | 0.313 | 0.323 | 0.359 |
| $\lambda = 0.9$ | Std. Dev. | | $5.18 \times 10^{-4}$ | 0.123 | 0.099 | 0.036 |
| | Std. Err. | | $4.10 \times 10^{-4}$ | 0.130 | 0.104 | 0.038 |
| | Est. | 0.499 | 0.305 | 0.302 | 0.320 | 0.461 |
| $\lambda = 0.95$ | Std. Dev. | | $6.15 \times 10^{-4}$ | 0.243 | 0.220 | 0.097 |
| | Std. Err. | | $6.02 \times 10^{-4}$ | 0.271 | 0.246 | 0.112 |

where $p(a|s)$ denote the probability that the system chooses policy $a$ at state $s$ and $Q_\cdot$ can be anyone of the Q-functions among $\{Q_q, Q_w, Q_{\text{mix}}\}$. Here, $Q_\cdot^{\text{REG}}(s, a)$ is an approximator of Q-funciton. The data-driven Q-functions are estimated by

$$\hat{Q}_{\cdot,j}^{\text{DR}}(a) = \hat{Q}_{\cdot,j}^{\text{REG}} + \frac{\mathbf{1}_{\{p=a\}}}{\hat{p}(a|s)}(\hat{Q}_{\cdot,j} - \hat{Q}_{\cdot,j}^{\text{REG}}), \tag{40}$$

where $\hat{Q}_\cdot$ has been introduced before and $\hat{p}$ is estimated by

$$\hat{p}(a = 0) = \frac{n_C}{n_C + n_T}, \ \hat{p}(a = 1) = \frac{n_T}{n_C + n_T}. \tag{41}$$

Next, we introduce the regressive approximation of the Q-functions. Since the estimated queue length $\hat{c}_q$ only returns marginal observation of the system, the Markov property is no longer kept well. Hence, we adopt a linear regression model that takes the past five costs as independent variables, i.e,

$$Q_j^{\text{REG}} = \beta + \sum_{u=0}^{4} \beta_u c_{q,j-u}. \tag{42}$$

Therefore, our regression is conducted on data set

$$\Big\{ (c_{q,j}, c_{q,j-1}, c_{q,j-2}, c_{q,j-3}, c_{q,j-4}; \hat{Q}_{\cdot,j}) : j \geq 4 \Big\}, \tag{43}$$

where $\cdot$ can be $q, w, mix$. Here we have only adopted queue-length-based costs as the independent variables because it is numerically the best among queue-length-based cost, response-time-based cost and mixed cost. Based on regression model (42), we construct parametric estimation of the Q-functions by

$$\hat{Q}_{\cdot,j}^{\text{REG}} = \hat{\beta} + \sum_{u=0}^{4} \hat{\beta}_u c_{\cdot,j-u}. \tag{44}$$

Substituting this into (40) leads to the data-driven estimation of doubly robust Q-functions. We further propose doubly robust Differences-in-Q estimators by

$$\widehat{DQ}_{\cdot}^{\text{DR}} = \frac{1}{n_C + n_T - 4 - L} \sum_{j=4}^{n_C + n_T - L} \left( \hat{Q}_{\cdot,j}^{\text{DR}}(1) - \hat{Q}_{\cdot,j}^{DR}(0) \right). \tag{45}$$

In essence, the doubly robust Q-functions represent a refinement in accurately estimating Q-functions. Beyond their inherent doubly robustness as discussed in the referenced literature Jiang and Li (2016), the addition of regressive estimation $\hat{Q}_{\cdot}^{\text{REG}}$ in (39) serves as a control variate, aiding in reducing variance in estimation. Unlike previous approaches outlined in Section 4.3, this control variate doesn't involve the interaction of average queue lengths and response times, thus sidestepping implications related to Little's Law. In Table 7 and Figure 6 below, we compare in pairs the queue-length-based Differences-in-Q estimator and the doubly robust queue-length-based Differences-in-Q estimator, the response-time-based Differences-in-Q estimator and the doubly robust response-time-based Differences-in-Q estimator, the mixed Differences-in-Q estimator and the doubly robust mixed Differences-in-Q estimator. Results from experiments indicate that doubly robust estimators notably reduce variance only when $\lambda$ is relatively large and the estimators don't involve mixed costs. On the other hand, when considering both response time and queue length concurrently, the doubly robust estimator exhibits minimal, if any, variance reduction effect. This observation underscores the significant role of Little's Law in comprehending and elucidating the dynamics of the queue system we consider.

Table 7: Doubly Robust Experiments with $d_1 = 3$, $d_2 = 2$

| $\lambda$ | Index | GTE | qDQ | qDQ-DR | wDQ | wDQ-DR | mixDQ | mixDQ-DR |
|---|---|---|---|---|---|---|---|---|
| | Est. | 0.138 | 0.147 | 0.146 | 0.140 | 0.140 | 0.138 | 0.138 |
| $\lambda = 0.5$ | Std. Dev. | | 0.030 | 0.030 | 0.010 | 0.009 | 0.004 | 0.004 |
| | Std. Err. | | 0.029 | 0.029 | 0.009 | 0.009 | 0.004 | 0.004 |
| | Est. | 0.187 | 0.180 | 0.180 | 0.183 | 0.183 | 0.184 | 0.184 |
| $\lambda = 0.6$ | Std. Dev. | | 0.032 | 0.031 | 0.013 | 0.013 | 0.006 | 0.006 |
| | Std. Err. | | 0.036 | 0.036 | 0.015 | 0.015 | 0.006 | 0.006 |
| | Est. | 0.252 | 0.242 | 0.243 | 0.246 | 0.246 | 0.249 | 0.249 |
| $\lambda = 0.7$ | Std. Dev. | | 0.043 | 0.043 | 0.023 | 0.023 | 0.009 | 0.008 |
| | Std. Err. | | 0.049 | 0.048 | 0.025 | 0.025 | 0.008 | 0.008 |
| | Est. | 0.358 | 0.348 | 0.347 | 0.348 | 0.347 | 0.348 | 0.348 |
| $\lambda = 0.8$ | Std. Dev. | | 0.073 | 0.071 | 0.047 | 0.046 | 0.015 | 0.015 |
| | Std. Err. | | 0.073 | 0.073 | 0.047 | 0.047 | 0.015 | 0.015 |
| | Est. | 0.439 | 0.421 | 0.415 | 0.423 | 0.419 | 0.428 | 0.428 |
| $\lambda = 0.85$ | Std. Dev. | | 0.094 | 0.087 | 0.068 | 0.063 | 0.022 | 0.022 |
| | Std. Err. | | 0.102 | 0.098 | 0.073 | 0.070 | 0.024 | 0.024 |
| | Est. | 0.566 | 0.485 | 0.480 | 0.497 | 0.493 | 0.541 | 0.541 |
| $\lambda = 0.9$ | Std. Dev. | | 0.138 | 0.117 | 0.110 | 0.094 | 0.044 | 0.044 |
| | Std. Err. | | 0.154 | 0.143 | 0.124 | 0.116 | 0.044 | 0.044 |
| | Est. | 0.797 | 0.422 | 0.445 | 0.459 | 0.480 | 0.730 | 0.732 |
| $\lambda = 0.95$ | Std. Dev. | | 0.299 | 0.192 | 0.268 | 0.175 | 0.114 | 0.114 |
| | Std. Err. | | 0.294 | 0.249 | 0.265 | 0.225 | 0.120 | 0.119 |

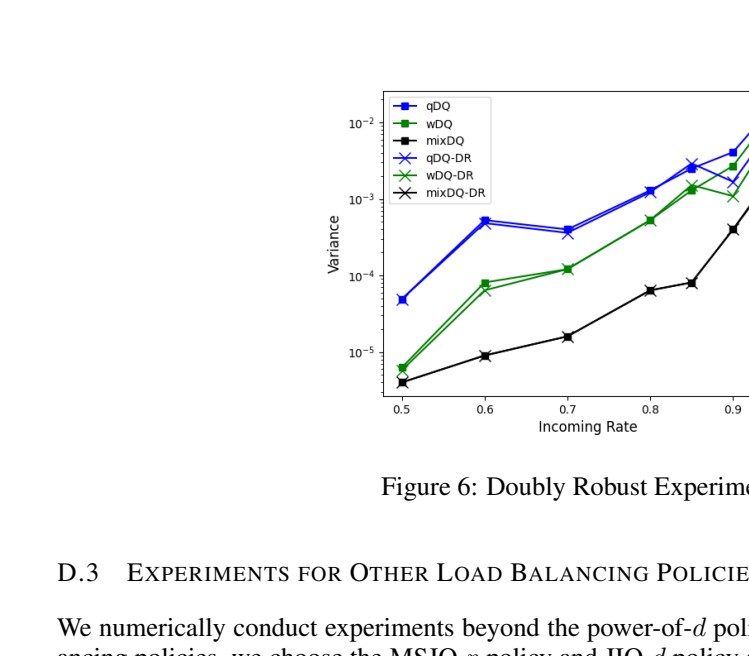

Figure 6: Doubly Robust Experiments

### D.3 EXPERIMENTS FOR OTHER LOAD BALANCING POLICIES

We numerically conduct experiments beyond the power-of-$d$ policy. Among all the other load balancing policies, we choose the MSJQ-$r$ policy and JIQ-$d$ policy as alternative illustrative examples to validate the Differences-in-Q theory. The MJSQ-$r$ policy, which originates from Banerjee et al. (2023), assigns new incoming jobs by the power-of-1 policy (uniformly and randomly to one of the servers in the system) with probability $r$ and by the power-of-$N$ policy (to the shortest server in the system) with probability $1-r$. We first present numerical results for the interference in MJSQ-$r_1/r_2$ policy, where $0 < r_1 < r_2 < 1$. Results are shown in Table 8 and Figure 7.

Table 8: MJSQ-$r_1/r_2$ Experiments with $r_1 = 0.4$, $r_2 = 0.6$

| $\lambda$ | Index | GTE | Naive | qDQ | wDQ | mixDQ |
|---|---|---|---|---|---|---|
| | Est. | 0.178 | 0.133 | 0.175 | 0.176 | 0.176 |
| $\lambda = 0.5$ | Std. Dev. | | $4.45 \times 10^{-4}$ | 0.036 | 0.012 | 0.006 |
| | Std. Err. | | $4.18 \times 10^{-4}$ | 0.035 | 0.012 | 0.006 |
| | Est. | 0.244 | 0.171 | 0.243 | 0.243 | 0.242 |
| $\lambda = 0.6$ | Std. Dev. | | $4.34 \times 10^{-4}$ | 0.039 | 0.015 | 0.008 |
| | Std. Err. | | $4.50 \times 10^{-4}$ | 0.040 | 0.016 | 0.008 |
| | Est. | 0.323 | 0.213 | 0.321 | 0.321 | 0.321 |
| $\lambda = 0.7$ | Std. Dev. | | $5.19 \times 10^{-4}$ | 0.044 | 0.020 | 0.010 |
| | Std. Err. | | $4.90 \times 10^{-4}$ | 0.048 | 0.022 | 0.011 |
| | Est. | 0.421 | 0.259 | 0.409 | 0.409 | 0.409 |
| $\lambda = 0.8$ | Std. Dev. | | $5.18 \times 10^{-4}$ | 0.061 | 0.035 | 0.016 |
| | Std. Err. | | $5.43 \times 10^{-4}$ | 0.065 | 0.037 | 0.016 |
| | Est. | 0.480 | 0.285 | 0.460 | 0.463 | 0.469 |
| $\lambda = 0.85$ | Std. Dev. | | $6.13 \times 10^{-4}$ | 0.085 | 0.056 | 0.025 |
| | Std. Err. | | $5.80 \times 10^{-4}$ | 0.083 | 0.055 | 0.024 |
| | Est. | 0.551 | 0.312 | 0.474 | 0.489 | 0.529 |
| $\lambda = 0.9$ | Std. Dev. | | $7.46 \times 10^{-4}$ | 0.110 | 0.086 | 0.044 |
| | Std. Err. | | $6.37 \times 10^{-4}$ | 0.124 | 0.095 | 0.047 |
| | Est. | 0.646 | 0.343 | 0.375 | 0.405 | 0.575 |
| $\lambda = 0.95$ | Std. Dev. | | $7.35 \times 10^{-4}$ | 0.266 | 0.236 | 0.128 |
| | Std. Err. | | $7.86 \times 10^{-4}$ | 0.260 | 0.234 | 0.140 |

Furthermore, we will conduct experiments on the MJSQ-$r$ policy and the power-of-$d$ policy. Here we set the MJSQ-$r$ policy as a control policy and the power-of-$d$ policy as a treatment policy. results are presented in Table 9 and Figure 8.

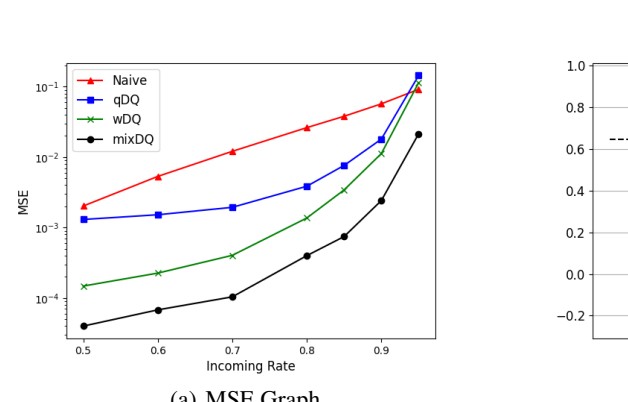

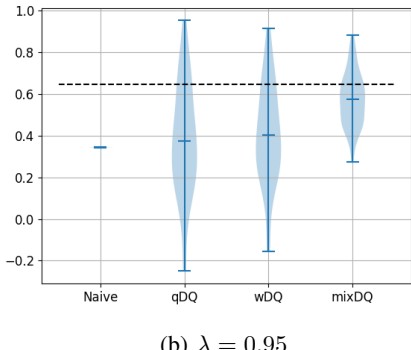

(a) MSE Graph

(b) $\lambda = 0.95$

Figure 7: MJSQ-0.4/0.6 Experiments

Table 9: MJSQ-$r$/Power-of-$d$ Experiments with $r = 0.95$, $d = 1$

| $\lambda$ | Index | GTE | Naive | qDQ | wDQ | mixDQ |
|---|---|---|---|---|---|---|
| $\lambda = 0.5$ | Est. | 0.095 | 0.049 | 0.101 | 0.098 | 0.095 |
| | Std. Dev. | | $7.54 \times 10^{-4}$ | 0.081 | 0.040 | 0.004 |
| | Std. Err. | | $8.60 \times 10^{-4}$ | 0.077 | 0.038 | 0.003 |
| $\lambda = 0.6$ | Est. | 0.174 | 0.072 | 0.159 | 0.165 | 0.173 |
| | Std. Dev. | | $1.10 \times 10^{-3}$ | 0.112 | 0.065 | 0.006 |
| | Std. Err. | | $1.06 \times 10^{-3}$ | 0.106 | 0.063 | 0.006 |
| $\lambda = 0.7$ | Est. | 0.349 | 0.110 | 0.293 | 0.310 | 0.348 |
| | Std. Dev. | | $1.39 \times 10^{-3}$ | 0.161 | 0.110 | 0.012 |
| | Std. Err. | | $1.40 \times 10^{-3}$ | 0.156 | 0.107 | 0.013 |
| $\lambda = 0.8$ | Est. | 0.826 | 0.181 | 0.433 | 0.518 | 0.818 |
| | Std. Dev. | | $1.87 \times 10^{-3}$ | 0.209 | 0.164 | 0.028 |
| | Std. Err. | | $2.01 \times 10^{-3}$ | 0.247 | 0.193 | 0.029 |
| $\lambda = 0.85$ | Est. | 1.447 | 0.246 | 0.528 | 0.682 | 1.416 |
| | Std. Dev. | | $2.67 \times 10^{-3}$ | 0.311 | 0.258 | 0.058 |
| | Std. Err. | | $2.58 \times 10^{-3}$ | 0.330 | 0.275 | 0.051 |
| $\lambda = 0.9$ | Est. | 3.034 | 0.360 | 0.711 | 0.982 | 2.871 |
| | Std. Dev. | | $4.16 \times 10^{-3}$ | 0.447 | 0.395 | 0.104 |
| | Std. Err. | | $3.63 \times 10^{-3}$ | 0.483 | 0.425 | 0.104 |
| $\lambda = 0.95$ | Est. | 9.346 | 0.622 | 0.646 | 1.201 | 7.580 |
| | Std. Dev. | | $6.83 \times 10^{-3}$ | 0.839 | 0.776 | 0.266 |
| | Std. Err. | | $6.02 \times 10^{-3}$ | 0.828 | 0.770 | 0.290 |

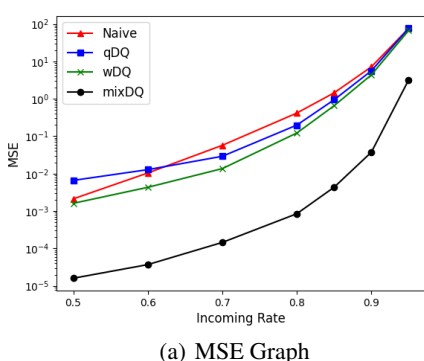

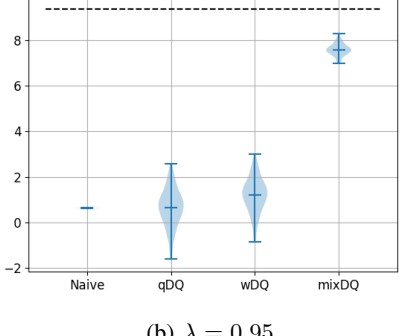

(a) MSE Graph         (b) $\lambda = 0.95$

Figure 8: MJSQ-0.95/Power-of-1 (random assignment) Experiments

Finally, we study the experiment related to the JIQ-$d$ policy. Initially, the JIQ policy was proposed in response to the heavy communication efforts caused by the power-of-$d$ policy. The JIQ policy assigns a new incoming job to one of the idle servers if there exists, and otherwise assigns the job randomly to one of the servers in the system. See Badonnel and Burgess (2008); Lu et al. (2011); Stolyar (2015) for more discussions on this scheduling policy. A straightforward generalization of the JIQ policy, proposed in Mukherjee et al. (2016b), is the so-called JIQ-$d$ policy, which assigns the new job to one of the idle servers, if there exists, and otherwise assigns the job according to the power-of-$d$ policy. We now present the results concerning JIQ-2/MJSQ-0.4 experiments in Table 10 and Figure 9.

Table 10: JIQ-$d$/MJSQ-$r$ Experiments with $d = 2$, $r = 0.4$

| $\lambda$ | Index | GTE | Naive | qDQ | wDQ | mixDQ |
|---|---|---|---|---|---|---|
| $\lambda = 0.5$ | Est. | 0.248 | 0.222 | 0.245 | 0.245 | 0.245 |
| | Std. Dev. | | $5.28 \times 10^{-4}$ | 0.024 | 0.005 | 0.004 |
| | Std. Err. | | $2.12 \times 10^{-4}$ | 0.023 | 0.005 | 0.004 |
| $\lambda = 0.6$ | Est. | 0.307 | 0.271 | 0.302 | 0.301 | 0.301 |
| | Std. Dev. | | $4.81 \times 10^{-4}$ | 0.027 | 0.007 | 0.005 |
| | Std. Err. | | $2.20 \times 10^{-4}$ | 0.024 | 0.007 | 0.005 |
| $\lambda = 0.7$ | Est. | 0.364 | 0.317 | 0.356 | 0.355 | 0.355 |
| | Std. Dev. | | $4.07 \times 10^{-4}$ | 0.030 | 0.011 | 0.007 |
| | Std. Err. | | $2.44 \times 10^{-4}$ | 0.030 | 0.012 | 0.007 |
| $\lambda = 0.8$ | Est. | 0.415 | 0.347 | 0.403 | 0.403 | 0.403 |
| | Std. Dev. | | $4.01 \times 10^{-4}$ | 0.045 | 0.027 | 0.015 |
| | Std. Err. | | $2.97 \times 10^{-4}$ | 0.047 | 0.028 | 0.015 |
| $\lambda = 0.85$ | Est. | 0.433 | 0.339 | 0.426 | 0.423 | 0.418 |
| | Std. Dev. | | $4.86 \times 10^{-4}$ | 0.075 | 0.053 | 0.025 |
| | Std. Err. | | $3.47 \times 10^{-4}$ | 0.069 | 0.048 | 0.025 |
| $\lambda = 0.9$ | Est. | 0.433 | 0.281 | 0.398 | 0.402 | 0.415 |
| | Std. Dev. | | $8.82 \times 10^{-4}$ | 0.118 | 0.094 | 0.044 |
| | Std. Err. | | $4.30 \times 10^{-4}$ | 0.113 | 0.091 | 0.048 |
| $\lambda = 0.95$ | Est. | 0.373 | 0.085 | 0.297 | 0.303 | 0.351 |
| | Std. Dev. | | $2.59 \times 10^{-3}$ | 0.280 | 0.254 | 0.135 |
| | Std. Err. | | $6.41 \times 10^{-4}$ | 0.258 | 0.236 | 0.137 |

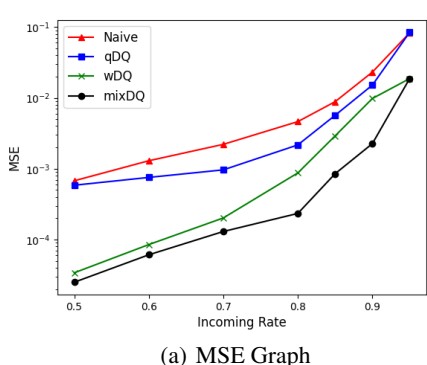

(a) MSE Graph

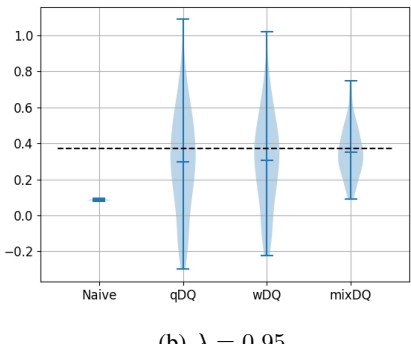

(b) $\lambda = 0.95$

Figure 9: JIQ-2/MJSQ-0.4 Experiments

### D.4 EXPERIMENTS WITH COMMUNICATION DELAY

In the most context of this paper, we have assumed the dispatcher to have super ability, i.e, the dispatcher can assign an incoming job to one of the servers according to some specific load balancing policies without delay. However, the information communication between the servers and the dispatcher may result in delay of job assignments. Below in this section, we present both the experiment design and numerical results for the Power-of-$d_1/d_2$ experiments with delay. Instead of adopting the periodic information update model from Mitzenmacher (1997), which assumes periodic information updates from the servers, we adopt an alternative random setting. Once a job arrives in the system, it will be stored in the dispatcher, and the d randomly chosen servers will be specified instantaneously. Here $d = d_1$ or $d_2$ depends on the polices chosen for this job. The job will wait at the dispatcher until all the $d$ servers send their queue information to the dispatcher. We suppose the delay time for the d servers are $q_1, q_2, \cdots, q_d$, where $\{q_1, q_2, \cdots, q_d\}$ are i.i.d. exponential random variables with unit rate. The total delay time for this job is defined by $\max\{q_1, q_2, \cdots, q_d\}$. After getting the queue information from the d servers, the job will be assigned to the shortest server and leaves the dispatcher.

To fit in the framework of Differences-in-Q estimators, we modify the definition of the response-time-based cost and the queue-length-based cost. Specifically, suppose a job arrives at the dispatcher at time $t$ when the dispatcher has exactly $n$ jobs and leaves the dispatcher at time $t'$. We make the following modifications to the cost functions:

$$\hat{c}_w \leftarrow (t' - t) + \hat{c}_w, \tag{46}$$

$$\hat{c}_q \leftarrow \frac{n}{N} + \hat{c}_q. \tag{47}$$

Then Q-functions and Differences-in-Q estimators are calculated as previous procedures. Below we present the numerical results in Table 11 and Figure 10.

### APPENDIX E   LLM USAGE

We use LLM purely for polishing writing.

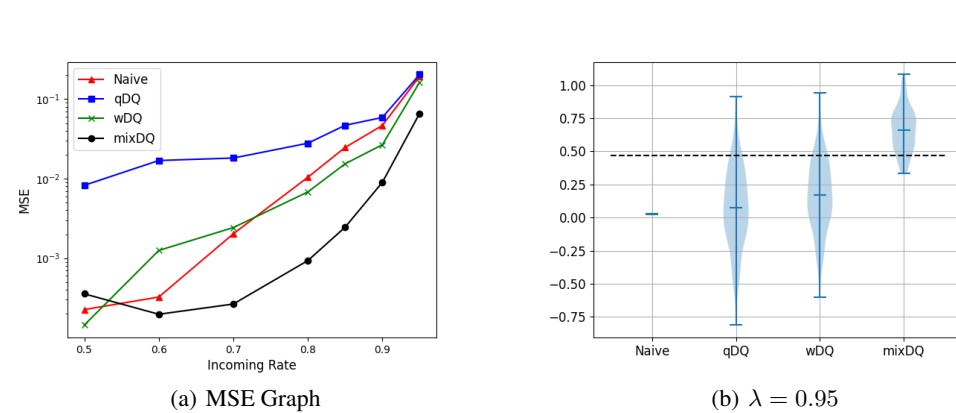

(a) MSE Graph                  (b) $\lambda = 0.95$

Figure 10: Power-of-3/2 Experiments with Delay

Table 11: Power-of-3/2 Experiments with Delay

| $\lambda$ | Index | GTE | Naive | qDQ | wDQ | mixDQ |
|---|---|---|---|---|---|---|
| | Est. | -0.216 | -0.201 | -0.297 | -0.216 | -0.200 |
| $\lambda = 0.5$ | Std. Dev. | | $7.62 \times 10^{-4}$ | 0.041 | 0.012 | 0.010 |
| | Std. Err. | | $7.69 \times 10^{-4}$ | 0.045 | 0.015 | 0.014 |
| | Est. | -0.147 | -0.165 | -0.269 | -0.178 | -0.148 |
| $\lambda = 0.6$ | Std. Dev. | | $6.88 \times 10^{-4}$ | 0.045 | 0.017 | 0.014 |
| | Std. Err. | | $7.24 \times 10^{-4}$ | 0.051 | 0.019 | 0.015 |
| | Est. | -0.080 | -0.125 | -0.203 | -0.123 | -0.077 |
| $\lambda = 0.7$ | Std. Dev. | | $7.18 \times 10^{-4}$ | 0.055 | 0.024 | 0.016 |
| | Std. Err. | | $6.98 \times 10^{-4}$ | 0.063 | 0.028 | 0.017 |
| | Est. | 0.023 | -0.079 | -0.117 | -0.041 | 0.039 |
| $\lambda = 0.8$ | Std. Dev. | | $7.35 \times 10^{-4}$ | 0.091 | 0.052 | 0.026 |
| | Std. Err. | | $6.97 \times 10^{-4}$ | 0.088 | 0.049 | 0.024 |
| | Est. | 0.106 | -0.051 | -0.070 | 0.013 | 0.141 |
| $\lambda = 0.85$ | Std. Dev. | | $7.73 \times 10^{-4}$ | 0.126 | 0.082 | 0.035 |
| | Std. Err. | | $7.12 \times 10^{-4}$ | 0.113 | 0.074 | 0.034 |
| | Est. | 0.233 | -0.017 | 0.044 | 0.119 | 0.308 |
| $\lambda = 0.9$ | Std. Dev. | | $7.43 \times 10^{-4}$ | 0.152 | 0.117 | 0.058 |
| | Std. Err. | | $7.51 \times 10^{-4}$ | 0.164 | 0.123 | 0.056 |
| | Est. | 0.466 | 0.026 | 0.131 | 0.167 | 0.660 |
| $\lambda = 0.95$ | Std. Dev. | | $1.02 \times 10^{-3}$ | 0.303 | 0.271 | 0.169 |
| | Std. Err. | | $8.82 \times 10^{-4}$ | 0.297 | 0.260 | 0.137 |

