# OpenReview forum: "Experimentation for Different Scheduling Policies on Queues: Mixed Differences-in-Q Estimators Based on Little’s Law"
_ICLR.cc/2026/Conference — Submitted to ICLR 2026_

### Official Review · Reviewer_Ef3v · 2025-10-21

**Soundness:** 2
**Presentation:** 2
**Contribution:** 2
**Rating:** 4
**Confidence:** 4

**Summary:**

This paper addresses the challenge of interference in A/B testing of scheduling policies in large-scale queuing systems, such as data centers. Standard A/B tests can produce biased estimates because treatment and control policies interact through shared system states (e.g., queue lengths).

The paper's main approach is to build on the Differences-in-Q (DQ) estimator of Farias et al. (2022), which reduces bias due to Markovian interference, the authors propose a mixed DQ estimator grounded in Little’s Law. The main contribution seems to be the idea of using two DQ estimators, one based on response time and one on queue length, and combine them (in a weighted way) to reduce variance. The paper provides some connections to Little’s Law. The authors perform extensive simulation results (focusing on variants of "choose the best of d queues" as policies, but adding variations to that basic setup, such as heterogeneous servers) to show that the mixed DQ estimator substantially reduces both bias and variance.

**Strengths:**

The new estimator appears to be a nontrivial extension of the past framework.

It's an interesting way to use LIttle's Law.

The control-variate formulation and the explicit derivation of the optimal mixing weight is provided well from a mathematical standpoint.

The simulation results are comprehensive: they cover variations such as service heterogeneity and communication delays.  The results from simulation are good.

**Weaknesses:**

The paper uses a lot of terminology and setup that would benefit from more explanation. For example, the paper defines the Global Treatment Effect (GTE) as the difference in long-run average performance (response time or queue length) between two policies. But it never clearly motivates why estimating this GTE matters in an experimental sense or its specific relation to A/B testing.

Similarly, the paper asserts that the DQ estimator “addresses Markovian interference,” but doesn't really explain how.  They seem to rely on the reader being familiar with Farias et al. (which I am not).  Markovian interference (temporal dependence between treatments) is apparently the motivation for DQ estimators, and  the authors assume the reader already knows how the DQ estimator corrects for it. They neither visualize nor walk through how Q-value differences remove bias from the evolving queue state.

(Said another way: Mixing two scheduling policies in one system creates interactions that do not correspond to how either policy performs alone -- that's the Markovian interference as I understand it.  But this makes it unclear, at least to me, what “ground truth” the estimators are meant to recover, especially under the interference that mixing creates. The paper needs a clearer explanation of what the mixed experiments are intended to demonstrate.)

The artificial setup of the power-of-d policies hurts the papers in various ways. First, because it's clear what the "right answer" is the impact is lessened.  Second, it's not clear why one would ever run a mixed systems (some jobs get d1 choices, some jobs get d2) and try to reverse engineer the right answer, rather than just run each system separately (which the authors do at some point in the paper).

The estimator’s variance minimization is derived under sample-based variance–covariance assumptions.

The truncation parameter L affects both bias and variance. The paper does not propose a principled selection rule.

All of the experiments are simulation-based. Even a small-scale real or trace-based evaluation (e.g., on a queueing simulator or an open-source cluster model) would strengthen the applied relevance.

**Questions:**

In the weaknesses I think are embedded a series of questions -- how do these estimators work?  Why are we looking at mixed systems, and why is it really OK to do so (particularly in this setting)?  How should L be chosen?  etc.

Maybe one point to make is that I got intrigued when I saw the appendix where communication delay was considered, because the power-of-d choices with communication delay has stranger properties -- in particular a smaller number of choices CAN be better because of "herd behavior" explained in the Mitzenmacher paper cited in that section.  I think that would have been a much better "test case" for the main paper, because the answer it not obvious.  (And because it complicates the question of how does the "mixed process" tell you about individual processes -- I'm not convinced that 2 choices doing better in a mixed process, for instance, means it would be better in an unmixed process when delay is involved!)

---

> ### Author Response · Authors · 2025-11-24
>
> We thank the reviewer for the valuable time and comments. Below we provide our responses to the Weaknesses and Questions raised.
>
> **W1. Motivation for GTE and mixed experiments.**
> In tech companies and many other operational environments, numerous A/B tests are conducted to estimate the performance difference between two competing policies. Knowing this difference—the GTE—is crucial for decision making, because the policy with better performance will ultimately be deployed system-wide. The GTE is the only causal estimand that answers the essential deployment question: “If we ran policy A globally, would the system improve?”
>
> In real data centers, A/B tests are implemented by randomly routing a fraction of jobs to one policy and the remaining fraction to the other. This mixed setting is not optional—it is exactly how real-world systems operate and how data are generated for evaluation.
>
> **W2. How DQ removes Markovian interference.**
> We apologize for not explaining clearly how DQ removes Markovian interference. Intuitively, the DQ estimators account not only for the immediate cost difference between the two policies, but also for all future downstream effects caused by the different state transitions that each policy induces. In other words, they capture both the direct reward difference and the subsequent impact of state evolution. This telescoping structure is what reduces the bias introduced by Markovian interference.  Farias et al. (2022) further show that the DQ estimator can significantly reduce the bias compared to the naive estimator. We have updated the manuscript to include this explanation.
>
>
> **W3. Running each system separately.**
> In practice, arrival dynamics are highly non-stationary, with task arrival rates fluctuating substantially over time. Moreover, the treatment effect (GTE) is typically much smaller than the day-to-day variability caused by this non-stationarity. As a result, running each policy separately would produce extremely high variance.
> In addition, deploying a new design to the entire system is risky and may cause operational harm if its performance is worse than expected. For these reasons, A/B tests with a mixed system—where only a fraction of traffic is routed to the new policy—are both necessary and standard practice in industry.
>
> **W4. The estimator’s variance minimization is derived under sample-based variance–covariance assumptions.**
> The sample-based variance–covariance estimates allow us to compute the mixing parameter $\hat{\alpha}$ directly from the data. Furthermore, the mixed estimator uses a standard control-variate construction. No extra assumptions or modeling are introduced: the covariance is empirically estimated exactly as in classical simulation variance-reduction methods.
>
> **W5. Truncation length $L$**
> We thank the reviewer for pointing out the lack of discussion regarding the choice of truncation length. We have added experiments with varying $L$, shown below.
>
> **Power-of-3/2 Experiments with $N=20$ and varying $L$.**
> GTE = 0.566. All entries are mean (std).
>
> | $\lambda = 0.9$     | Naive | qDQ            | wDQ            | mixDQ          |
> |------------------------|-------|----------------|----------------|----------------|
> | $L = 10N\lambda$     | 0.316 | 0.391 (0.039)  | 0.404 (0.030)  | 0.537 (0.012)  |
> | $L = 30N\lambda$    | 0.316 | 0.462 (0.067)  | 0.479 (0.054)  | 0.544 (0.021)  |
> | $L = 60N\lambda$     | 0.316 | 0.533 (0.118)  | 0.537 (0.097)  | 0.550 (0.033)  |
> | $L = 100N\lambda$    | 0.316 | 0.557 (0.139)  | 0.556 (0.114)  | 0.551 (0.046)  |
>
> Under the same experimental settings, we observe a clear bias-variance trade-off. When the truncation length $L$ is small, the variance is typically lower, but the Q-functions suffer from bias due to insufficient trajectory length. Conversely, when $L$ is large, the Q-function estimates converge better, but the variance increases significantly, which limits the estimator's performance.  However, the superiority of our method is not sensitive to the particular choice of $L$. Additionally, we find that the mixed Differences-in-Q estimator converges faster than its counterparts when $L$ is small. This further demonstrates the effectiveness of our proposed method.
>
> **Question: communication delay**
> We agree with the reviewer that a smaller number of choices can be better due to “herd behavior.” However, our paper is not concerned with designing or identifying the optimal policy. Our goal is to estimate the performance difference between two given policies in an A/B test setting. We also do not claim that the two-choice policy is better.
>
> By adjusting parameters, one policy may outperform the other in some regimes, while the opposite may hold in different regimes. This variability is exactly why industry relies on A/B testing: theoretical analysis alone cannot definitively determine which policy will perform better under real operational conditions.

---

> > ### Comment · Reviewer_Ef3v · 2025-11-25
> >
> > I have seen the responses given by the authors to my and other reviews.  I appreciate their work.  I am planning to keep my score the same.

---

> > > ### Author Response · Authors · 2025-11-26
> > >
> > > We hope that we have adequately addressed the key concerns regarding (i) why companies in practice rely heavily on A/B tests to estimate policy performance differences, (ii) why running each system separately is typically infeasible due to non-stationarity, and (iii) the intuition behind the construction of the DQ estimator.
> > >
> > > If you feel that your comments have been fully addressed and that our work is now clearer and more compelling, we would greatly appreciate it if you could kindly reconsider your score.
> > >
> > > Thank you very much for your time and thoughtful feedback.

---

### Official Review · Reviewer_eqJd · 2025-10-31

**Soundness:** 2
**Presentation:** 3
**Contribution:** 3
**Rating:** 4
**Confidence:** 4

**Summary:**

The paper studies A/B testing for load-balancing (scheduling) policies in parallel-server queueing systems, where naive A/B comparisons suffer from Markovian / intertemporal interference. Building on the Differences-in-Q (DQ) estimator of Vivek Farias et al. (2022), the authors introduce a mixed Differences-in-Q estimator that combines a response-time-based Q and a queue-length-based Q using weights motivated by Little’s Law (L = λW). They derive an explicit variance expression for the mixed estimator and an analytic optimal weight α* (estimated from data) that minimizes variance; they describe estimators for the required variance/covariance terms and practical implementation details (including estimating λ and µ_i), and present extensive simulation studies (homogeneous/heterogeneous services, non-stationary arrival rates, delayed communication, power-of-d policies and others) showing markedly reduced MSE relative to naive and to single-type DQ estimators. The paper also discusses doubly-robust extensions and provides inference heuristics.

**Strengths:**

The paper presents a variance expression and an algebraic optimal weight α*, but does not give rigorous statistical guarantees for finite samples or for the consistency/convergence of an estimate of α̂ to α*. The Authors rely mainly on heuristic/sample formulas and Monte Carlo evidence; there is no main result (theorem) for basic properties of the estimator, like its consistency, asymptotically normality, or even that α̂ is well-behaved under the mixing experiment design. So, this seems to be a major concern; when the key aim/claim is variance reduction.


The truncation parameter (L) is used heavily and is admitted to cause non-negligible truncation bias, yet the paper gives minimal formal analysis of truncation error or principled rules to choose (L).

====

Presentation: Overall the exposition is clear and the paper is well organized (intro, the problem setting, definitions of estimators, mixing derivation, implementation, comprehensive experiments, appendices with proofs). Figures and Tables (e.g., tables for MSE/Std Dev across λ values, violin plots) support the running text.


Variance and SE estimates used for inference are heuristic (Section C.3). There is no careful finite-sample coverage study (e.g., empirical CI coverage over replications is not reported), so the computed CIs may be misleading.

=====

Contribution: This paper makes a solid, original contribution at the intersection of experimental design for systems and statistical estimation:

Conceptual novelty: the idea to mix queue-length and response-time Q-estimates via Little’s Law (a domain-informed control variate) is simple, elegant, and novel in the context of DQ estimation for Markovian interference.


Some other positives of the work: the estimator is easy to implement, leverages observables in typical dispatchers, and substantially reduces MSE in diverse, realistic scenarios—making it attractive to practitioners who run online experiments in data centers.


Some parts of the empirical computations are through, though: wide experiment coverage (heterogeneity, delays, non-stationarity) and ablations (power-of-5/3, doubly-robust variants) support the robustness of the proposed scheme.

**Weaknesses:**

The paper considers M/M/1 queue; this is a good model to start with. Perhaps, some other service times could have also been considered (M/G/1 queue).

The chosen heuristic (L=\lfloor 30 N\lambda\rfloor) appears a bit ad hoc; the empirical advantage of mixDQ may hinge on this choice (and the direction of truncation bias seems to help the mixed estimator in some regimes). So, it is not clear how the proposed mixed estimator is generally better.

The inference/SE scheme (Section C.3) uses a simple variance estimator and asymptotic CI; some more discussion about finite-sample validity or bootstrap alternatives would help.

A few places could be improved for clarity and reproducibility: The discussion of truncation bias (choice of L and its impact) is informative but could be expanded (guidance on choosing L in practice, sensitivity plots).

A scholarly survey is by Ward Whitt; among others, it offers a sample path based proof (a minor correction was provided later). A bit surprising that it is not referenced.

A major comment: there are _glarring_ typos in the references, including names of Authors of the papers. The first formal proof of Little's law is due to John D.C. Little (the paper is about this relation; the result has been used since decades before, we were told); a reference lists `Little and D. C. John'.

Another one is `OR', not `Or'. And similar ones.

**Questions:**

*) How sensitive are the results to the truncation length L? Can the Authors conduct a small extra computational experiment that brings out  the bias/variance trade-off (or MSE) as L varies (for at least one λ setting)?


*) As this paper considers M/M/1 queue, perhaps, the Authors comment on M/G/1 queue.

*) Some comments on the per-arrival computational/memory overhead to compute Q̂mix online would be useful. Can the method be implemented in streaming systems with low latency?

*) Typos listed above can (and should) be addressed.

---

> ### Author Response · Authors · 2025-11-24
>
> We thank the reviewer for the valuable time and comments. Below we provide our responses to the Weaknesses and Questions raised.
>
> **Q/W1: Regarding the truncation length $L$.**
> We thank the reviewer for pointing out the lack of discussion regarding the choice of truncation length. We have added experiments with varying $L$, shown below.
>
> **Power-of-3/2 Experiments with $N=20$ and varying $L$.**
> GTE = 0.566. All entries are mean (std).
>
> | $\lambda = 0.9$     | Naive | qDQ            | wDQ            | mixDQ          |
> |------------------------|-------|----------------|----------------|----------------|
> | $L = 10N\lambda$     | 0.316 | 0.391 (0.039)  | 0.404 (0.030)  | 0.537 (0.012)  |
> | $L = 30N\lambda$    | 0.316 | 0.462 (0.067)  | 0.479 (0.054)  | 0.544 (0.021)  |
> | $L = 60N\lambda$     | 0.316 | 0.533 (0.118)  | 0.537 (0.097)  | 0.550 (0.033)  |
> | $L = 100N\lambda$    | 0.316 | 0.557 (0.139)  | 0.556 (0.114)  | 0.551 (0.046)  |
>
> Under the same experimental settings, we observe a clear bias-variance trade-off. When the truncation length $L$ is small, the variance is typically lower, but the Q-functions suffer from bias due to insufficient trajectory length. Conversely, when $L$ is large, the Q-function estimates converge better, but the variance increases significantly, which limits the estimator's performance.  However, the superiority of our method is not sensitive to the particular choice of $L$. Additionally, we find that the mixed Differences-in-Q estimator converges faster than its counterparts when $L$ is small. This further demonstrates the effectiveness of our proposed method.
>
> **Q/W2: Regarding the M/G/1 experiments.**
> We thank the reviewer for raising the general-service-time case. In addition to the exponential service-time experiments in the manuscript, we include the following experiment where the service time is constant and equal to the inverse of the service rate. This serves as a representative example of a general (non-exponential) service-time distribution.
>
> **Power-of-3/2 experiments in the constant-service-time setting**
>
> | $\lambda$ | Index      | GTE    | Naive              | qDQ      | wDQ      | mixDQ    |
> |-------------|------------|--------|--------------------|----------|----------|----------|
> | **0.85**    | Est.       | 0.201  | 0.132              | 0.191    | 0.193    | **0.196** |
> |             | Std. Dev.  |        | $2.78 \times 10^{-4}$ | 0.041    | 0.026    | 0.009    |
> | **0.9**     | Est.       | 0.261  | 0.149              | 0.243    | 0.245    | **0.251** |
> |             | Std. Dev.  |        | $2.87 \times 10^{-4}$ | 0.070    | 0.051    | 0.017    |
> | **0.95**    | Est.       | 0.375  | 0.170              | 0.278    | 0.288    | **0.348** |
> |             | Std. Dev.  |        | $4.10\times10^{-4}$ | 0.145    | 0.127    | 0.052    |
>
> *Notes:*
> - In the *Index* column, **Est.** = *Estimator* and **Std. Dev.** = *Standard Deviation*.
> - **qDQ**, **wDQ**, and **mixDQ** refer to the queue-length–based, response-time–based, and mixed DQ estimators, respectively.
>
>
> These results align with our previous findings and show that the mixed Differences-in-Q estimator continues to achieve substantial variance reduction in the M/G/1 setting.
>
> **Q/W3: Regarding finite-sample validity or bootstrap alternatives.**
> We agree that statistical inference under interference is challenging, as emphasized by Farias et al. A practical remedy is to run A/A tests to estimate the variance. Our experiments show that the variance estimated from A/A tests typically provides a reliable approximation of the variance observed in the corresponding A/B tests.
>
> **Q/W4: Regarding presentation.**
> We appreciate the reviewer for identifying typos and insufficient citations. We are sorry to misspell the name of John D.C. Little in one of the citations. We have corrected these issues in the updated manuscript.
>
> We thank the reviewer again for the thoughtful feedback and constructive comments.

---

### Official Review · Reviewer_z6A5 · 2025-11-01

**Soundness:** 3
**Presentation:** 3
**Contribution:** 2
**Rating:** 4
**Confidence:** 2

**Summary:**

The paper studies experimentation for evaluating scheduling policies in queueing systems (e.g., data center) under Markovian interference, where the performance of one policy can affect that of another through shared system states. Building on the Differences-in-Q estimator proposed by Farias et al. (2022), the authors introduce a mixed Differences-in-Q estimator that leverages Little’s Law to combine queue-length–based and response-time–based estimators. This method is shown theoretically and empirically to reduce the estimator’s variance while maintaining low bias. The paper provides detailed simulation studies across a variety of settings—including heterogeneous service rates, non-stationary arrivals, and communication delays—to demonstrate robustness and accuracy.

**Strengths:**

The paper extends the Differences-in-Q estimator to a queueing context through an elegant use of Little’s Law, yielding a mixed estimator with improved bias–variance performance. The idea of combining queue-length–based and response-time–based estimators via an analytically derived mixing weight is original within the experimentation and queueing literature, and the empirical evaluation is extensive, systematic, and convincingly supports the claims. The work is clearly motivated by real-world data-center scheduling and includes comprehensive numerical experiments. Overall, this paper effectively bridges causal inference methodology with queueing theory and offers practical insight into experimentation for dynamic systems.

**Weaknesses:**

While the paper is technically sound and makes a meaningful queueing-theoretic contribution, its core novelty lies primarily in the application of Little’s Law to construct a variance-reduced estimator, rather than in advancing causal inference or learning methodology. As such, the work reads more as an operations research study on experimental evaluation in queueing systems than as a contribution to causal reasoning or machine learning. In contrast, the Differences-in-Q estimator by Farias et al. (NeurIPS 2022) was framed as a general method for causal inference under Markovian interference—an issue broadly relevant to ML experimentation and policy evaluation—whereas the current paper focuses specifically on queuing systems.

**Questions:**

To strengthen its fit for ICLR, the authors could better articulate how the proposed estimator generalizes beyond queueing contexts or connects to broader themes such as off-policy evaluation, dynamic treatment effects, or learning under interference.

1. Could the authors elaborate on whether the Little’s Law–based variance reduction idea can extend to other Markovian environments or experimental settings with temporal interference?

2. How does the proposed estimator relate conceptually to methods in off-policy evaluation or causal inference under dependent data? For example, could it be interpreted as a form of variance reduction or control variate in the context of policy evaluation?

3. Do the authors see a pathway for connecting their approach more directly to broader machine learning themes such as dynamic treatment effects, sequential experimentation, or learning under interference?

---

> ### Author Response · Authors · 2025-11-24
>
> We thank the reviewer for the valuable time and comments. Below we provide our responses to the Weaknesses and Questions raised.
>
> **Weakness: contribution over Farias et al. (NeurIPS 2022)**
>  We agree with the reviewer that Farias et al. (NeurIPS 2022) provide a powerful approach to interference in general causal inference settings. However, directly applying their method to queueing systems, which are central in machine learning and have numerous applications, faces significant challenges. As shown in our work, Differences-in-Q estimators typically suffer from very large variance in queueing environments. This high variance severely limits the usefulness of such estimators for causal inference. Our method is motivated by this observation. By leveraging domain knowledge specific to queueing systems—most notably Little’s Law—we construct mixed Differences-in-Q estimators that achieve substantial variance reduction and yield large improvements in mean squared error. Moreover, because Little’s Law is a fundamental property of virtually all queueing systems, our approach applies broadly to any queueing environment, including but not limited to data-center workloads.
>
> **Q1. Little’s Law**
> Little’s Law is a domain-specific structural property of queueing systems and applies far more broadly than the supermarket model. Consequently, our approach extends to a wide range of queueing environments. More importantly, our results illustrate how domain knowledge can be systematically exploited for variance reduction in queueing-based causal inference. For other Markovian systems beyond queueing, similar gains may be achievable by leveraging their own structural properties.
>
>
> **Q2. Off-policy evaluation or causal inference under dependent data**
> Yes. Our mixed Differences-in-Q estimator can indeed be viewed as a control-variate–based variance-reduction method, as discussed on Line 287 in Section 4.3. Our experiments show that its variance is significantly smaller than that of standard Differences-in-Q estimators. In addition, Farias et al. demonstrate that off-policy estimators can suffer from extremely large variance in Markovian settings. This further motivates our variance-reduction approach.
>
> **Q3. Connection to broader ML themes such as dynamic treatment effects, sequential experimentation, or learning under interference**
> Our work focuses on causal inference under interference. By leveraging Little’s Law, an intrinsic property of queueing systems, we target the estimation of global treatment effects under interference. We also believe the insights developed here may benefit reinforcement learning, sequential experimentation, and other areas where interference and high variance are major challenges. For example, in reinforcement learning for queueing systems, one may learn two value models: one using average waiting time as the reward and another using average queue length. By optimally combining these two estimators, one can achieve significant variance reduction.
>
> We thank the reviewer again for your inspiring comments and great efforts!

---

> > ### Comment · Reviewer_z6A5 · 2025-11-26
> >
> > Thanks for the authors' clarification. I can see the value of this work. If you can improve the manuscript to improve its generality (e.g., rather than data centers and supermarket model in the contribution, mention broader applications and frameworks), I'm happy to raise the scores.

---

> > > ### Author Response · Authors · 2025-11-28
> > >
> > > Thanks for your suggestion. We have updated our contributions in the introduction and the conclusion. Specifically, we have added a discussion on how incorporating domain knowledge can improve evaluation, and we have outlined several future directions—such as leveraging structural properties in inventory control and dynamic pricing—to develop more decision-focused evaluation metrics. We also copied our updates below.
> > >
> > > >Finally, our findings highlight a broader insight: incorporating domain knowledge—specifically,
> > > the structural properties of queueing systems—can substantially reduce variance and improve the
> > > reliability of performance evaluation.
> > >
> > > >More broadly, we plan to extend this idea of leveraging domain knowledge to improve evaluation methodologies. For example, in assessing demand-prediction models for inventory control, we aim to exploit the well-known optimality of the
> > > $s,S$ policy and develop a decision-focused scoring rule—grounded in the newsvendor-type loss—to more accurately evaluate prediction quality in terms of operational performance.
> > > Furthermore, for applications such as dynamic pricing, we plan to exploit structural properties—such as revenue concavity, value-function convexity, and price monotonicity—to design decision-focused evaluation metrics that more directly capture how prediction errors affect pricing performance.

---

### Official Review · Reviewer_2Fjs · 2025-11-03

**Soundness:** 3
**Presentation:** 3
**Contribution:** 3
**Rating:** 6
**Confidence:** 4

**Summary:**

The paper studies A/B testing for queueing-based scheduling under Markovian interference and proposes a mixed Differences-in-Q (DQ) estimator that blends a response-time DQ with a queue-length DQ using Little’s Law. They show response-time and queue-length GTEs are equivalent up to λ (Proposition 1), then derive an optimal mixing weight α via a control-variate variance calculation, and validate on supermarket-style load balancing with heterogeneous service, non-stationary arrivals, and delays.

**Strengths:**

1. The mixing idea is indeed interesting: Blending two DQs linked by Little’s Law and choosing α to minimize the mixed Q-variance is neat and practically motivated. The paper explicitly derives Var(Q̂_mix(α)) and the closed-form α* (Eqs. 15–16), then shows large variance drops versus plain DQ in experiments.

2. Breadth of experiments: They cover homogeneous/heterogeneous service, non-stationary λ(t), policy families beyond power-of-d, doubly-robust approaches of estimating Q, and communication delay; mixed-DQ tends to achieve the lowest MSE.

**Weaknesses:**

1. Any analysis for the mixing idea rather than the empirical correlation? – You do set up a principled control-variate estimator and give a closed-form α* (Eq. 16). That’s good. But it would be nice to have some intuition for the strong correlation between these two Q-functions.

2. Any value decomposition trick to reduce dimensionality? For estimating the Q function of Q(q1, q2, ... qN), one trick is to estimate Q(q1), Q(q2), ..., Q(qN) and add them up, so-called value decomposition trick. Have you considered this approximation in the doubly-robust estimation?

3. In addition, have you considered estimating V-value function and use Monte-carlo to estimate the advantage function (similar to what is used in TRPO), usually that is a good trick for variance reduction in advantage function/Q function estimation

4. The motivation for this problem can be further highlighted: in data center, why can't we run a switchback experiment for this?

**Questions:**

See the weakness above.

---

> ### Author Response · Authors · 2025-11-24
>
> We sincerely thank the reviewer for the time and thoughtful feedback. Below we provide our point-by-point responses to the four weaknesses raised.
>
> **W1: Strong correlations between Q functions.**
>  We here provide an intuitive explanation for the strong correlation between the two Q-functions as follows. As the number of queues $N$ grows large, the stochastic queueing system converges to a deterministic ODE limit [1]:
> $\frac{d s_{i}}{d t} =\lambda  \left(ps_{i-1}^{d_{2}}+(1-p)s_{i-1}^{d_{1}} -ps_{i}^{d_{2}}-(1-p)s_{i}^{d_{1}}\right) -\left(s_{i} -s_{i +1}\right),  i=1,2,\cdots$.
> In this regime, the two Q-functions converge to their deterministic counterparts in the ODE system. One can show that these limiting Q-functions coincide. Because the finite-N Q-functions concentrate around these common limits, they inherit a strong correlation. This shared structure explains why our mixed estimator achieves substantial variance reduction.
>
> **W2: Value decomposition.**
> We have experimented with value decomposition. Empirically, we find that it does not substantially improve the performance of the doubly-robust Differences-in-Q estimators. Under the experimental settings in Table 7 with \(\lambda = 0.9\), our current doubly-robust waiting-time-based Differences-in-Q estimator achieves a standard deviation of \(0.094\), whereas value decomposition yields a standard deviation of \(0.090\). The limited gain is primarily due to substantial bias in the estimated Q-functions. In our experiments, value-decomposition-based estimations tend to be largely biased. A deeper investigation into the structural patterns of Q-functions in queueing systems is an interesting direction in future work.
>
> **W3: Value decomposition.**
>  We agree that using advantage functions could further reduce variance. However, estimating V-functions in our setting also relies on Monte Carlo rollouts as the underlying states could be very complicated, and alternative estimation methods introduce significant bias in practice. Given this point, we choose the current approach for stable and reliable inference.
>
> **W4: Why not switchback experiments.**
>  We thank the reviewer for raising this point. In practice, arrival dynamics are highly non-stationary, with task arrival rates varying substantially over time. As a result, switchback designs would induce large variance. Given this variance amplification, together with the already high variance of Q-function estimates, we believe that a fully randomized experimental design is a more reliable approach for comparing scheduling policies.
>
> We again thank the reviewer for the constructive comments and careful consideration!
>
> [1] Mitzenmacher, M., 2002. The power of two choices in randomized load balancing. IEEE Transactions on Parallel and Distributed Systems, 12(10), pp.1094-1104.

---

> > ### Comment · Reviewer_2Fjs · 2025-11-27
> >
> > The authors have addressed my comments satisfactorily. I also appreciate the additional experiments on value decompositions. Overall, this is an interesting paper that introduces useful techniques for Q-value estimation for the queueing community, with potential impact beyond causal inference to control and reinforcement learning more broadly. I have raised my score.

---

### Author Response · Authors · 2025-11-24

Dear Reviewers,

We sincerely thank you for your thoughtful and constructive comments. They have greatly helped us improve the paper. In the revised manuscript, we have added two new experiments:

**1.General service-time distribution:** We include experiments with a general service-time distribution in Section 5.4, which further validates the robustness and performance of our methods in broader settings.

**2.Ablation on truncation length:** We conduct an ablation study on different choices of truncation length in Section 5.5 and confirm that our results are not sensitive to this parameter.

Best wishes,
Authors of Submission 8519

---

### Author Response · Authors · 2025-12-01

Dear Reviewers and Area Chair,

We appreciate the opportunity to present our contributions and the substantial improvements made during the rebuttal period.

## Summary of Contributions
This manuscript develops a framework for incorporating domain knowledge into experimental design, with a focus on A/B testing for scheduling policies in data centers. Building on the DQ estimators of Farias et al. (2022) and motivated by Little’s Law, we use both response time and average queue length as cost functions. To estimate the Global Treatment Effect (GTE), we introduce the mixed Differences-in-Q (DQ) estimator, which integrates the two cost measures to simultaneously reduce bias and variance. Little’s Law provides the theoretical foundation for this combination.

We conduct extensive simulations across heterogeneous service rates, non-stationary arrivals, and communication delays, and we evaluate a broad collection of scheduling policies. Our numerical results show that the proposed A/B testing framework consistently achieves low bias and low variance across a wide range of practical settings.

## Rebuttal Summary
In response to the reviewer feedback, we strengthened both the empirical evidence and the presentation. The main updates are:

1. **Additional experiments.**
   We added two new sets of experiments.
   - **Section 5.4:** We replace exponential service times with fixed service times, a representative case of general service-time distributions. The method continues to exhibit low bias and variance.
   - **Section 5.5:** We examine the impact of varying truncation lengths \(L\). The results reveal a clear bias–variance trade-off: small \(L\) yields low variance but higher bias due to truncated Monte Carlo Q-function estimates, while large \(L\) reduces bias but increases variance. However, the superiority of our method is not sensitive to the particular choice of $L$.

2. **Improved presentation.**
   We corrected typographical errors, updated citations, and added relevant references for clarity and context. We also expanded the intuition behind why DQ estimators are effective.

3. **Polished contributions and broader impact.**
   The revised manuscript now more clearly explains how Little’s Law—our central piece of domain knowledge—strengthens estimator performance. We also discuss applications beyond queueing systems, including inventory management, dynamic pricing, and online platforms, underscoring the broader relevance of our idea.

---

### Meta-Review · Area_Chair_9C6Z · 2026-01-05

**Summary:**

Paper is clear and well written, and the idea of mixing two Differences-in-Q estimators with Little’s Law is neat for queueing systems. Experiments are broad and realistic. But main concerns stay: contribution feels narrow and mostly applying known queueing theory (Little’s Law) rather than new ideas for causal inference or ML. Method depends on heuristics like truncation length and variance estimation, with no strong theory or guarantees. Rebuttal added experiments and clarified things, but didn’t fix the lack of theoretical grounding or generality beyond queueing. Overall, solid work but looks more like applied OR than an ICLR paper, hence we recommend reject.

**Reviewer Concerns:**

Rebuttal helped a bit: more experiments, better explanations, fixed presentation issues. Still big gaps: no formal guarantees for the mixed estimator, inference and variance steps are heuristic, novelty outside queueing is thin. Questions on truncation choice and ML relevance only partly answered.

**Reviewer Scores:**

Positive-but-cautious reviewers might stay the same or slightly up. Those worried about theory and scope will keep low scores. Overall, distribution still mixed and below the bar.

---

### Decision · Program_Chairs · 2026-01-26

Reject